# A graph-based cell tracking algorithm with few manually tunable parameters and automated segmentation error correction

Katharina Löffler[1,2]*, Tim Scherr[1], Ralf Mikut[1]

**1** Institute for Automation and Applied Informatics, Karlsruhe Institute of Technology, Eggenstein-Leopoldshafen, Germany, **2** Institute of Biological and Chemical Systems - Biological Information Processing, Karlsruhe Institute of Technology, Eggenstein-Leopoldshafen, Germany

* katharina.loeffler@kit.edu

**Data Availability Statement:** The Supporting information file "S1 Data Availability" describes how to reproduce the results of our tracking algorithm, and we have published all tracking

## Abstract

Automatic cell segmentation and tracking enables to gain quantitative insights into the processes driving cell migration. To investigate new data with minimal manual effort, cell tracking algorithms should be easy to apply and reduce manual curation time by providing automatic correction of segmentation errors. Current cell tracking algorithms, however, are either easy to apply to new data sets but lack automatic segmentation error correction, or have a vast set of parameters that needs either manual tuning or annotated data for parameter tuning. In this work, we propose a tracking algorithm with only few manually tunable parameters and automatic segmentation error correction. Moreover, no training data is needed. We compare the performance of our approach to three well-performing tracking algorithms from the Cell Tracking Challenge on data sets with simulated, degraded segmentation—including false negatives, over- and under-segmentation errors. Our tracking algorithm can correct false negatives, over- and under-segmentation errors as well as a mixture of the aforementioned segmentation errors. On data sets with under-segmentation errors or a mixture of segmentation errors our approach performs best. Moreover, without requiring additional manual tuning, our approach ranks several times in the top 3 on the 6th edition of the Cell Tracking Challenge.

## Introduction

The ability of cells to migrate is essential for many biological processes such as tissue formation, immune response, or wound healing [1]. Disruptions in cell migration can contribute to diseases such as malformation [2], autoimmune disease [3], and metastasis [4]. To better understand the mechanisms driving cell migration, the cell behavior can be analyzed quantitatively, for instance by tracking cells over time. However, tracking cells manually is tedious, even for small data sets, and becomes for large data sets infeasible. Therefore, automated cell tracking methods are needed which minimize manual curation effort and expert knowledge for parameter adjustments.

results from our analysis on Zenodo (https://zenodo.org/record/5227610 and https://zenodo.org/record/5227595).

**Funding:** We are grateful for funding by the Helmholtz Association in the program Natural, Artificial and Cognitive Information Processing (TS, RM) and HIDSS4Health - the Helmholtz Information & Data Science School for Health (KL, RM). The funders had no role in study design, data collection and analysis, decision to publish, or preparation of the manuscript. We acknowledge support by the KIT-Publication Fund of the Karlsruhe Institute of Technology.

**Competing interests:** The authors have declared that no competing interests exist.

Recent cell tracking methods can be categorized into tracking by detection and tracking by model evolution approaches [5]. In tracking by model evolution approaches an initial segmentation is propagated over time [6], whereas tracking by detection approaches split segmentation and tracking in two steps. In this paper, we focus on tracking by detection approaches, due to promising improvements of cell segmentation algorithms [7–11].

At present a vast variety of tracking by detection approaches has been proposed. The most simplistic approaches use nearest neighbor methods [12, 13] or are based on overlap [14, 15]. Bayesian filters like the Kalman filter [16], particle filter [17–19] or Bernoulli filter [20, 21] have been adapted for cell tracking as well. Hybrid methods combine simplistic tracking methods, like nearest neighbors, with more sophisticated tracking methods [22–25]. Furthermore, deep learning based approaches have been proposed for cell tracking [26, 27]. Graph-based approaches offer the possibility to model cell behavior such as motion, mitosis or cell death explicitly [28–36].

Tracks are created by linking segmentation masks over time based on a "linking" measure. A simple linking measure is the Euclidean distance between the positions of the cell centroids. Other linking measures are based on handcrafted features, such as position and appearance [25, 30, 37, 38], features of the cell's neighborhood [39], features derived from a graph structure [40], or learned features [18, 19, 26, 27, 41]. The contribution of the extracted features in the measure is often learned for instance by using logistic regression [30], a structured support vector machine [42], a random forest [33] or training convolutional neural networks [18, 19]. Besides that, some approaches train additional detectors to detect mitosis [12, 14, 19]. However, to fit such linking measures to new data sets, annotated data are needed, which requires additional annotation effort. Using simple, position-based linking measures, in contrast, can be applied to a vast set of experiments without training.

As tracking by detection methods split tracking and segmentation, a reasonable segmentation quality of cells is needed for good tracking results. However, segmentation approaches need to handle challenging imaging conditions such as low signal-to-noise ratio or low contrast [6] as well as the wide range of appearance due to different imaging methods and cell types [5]. Therefore, an error-free segmentation is almost impossible. The resulting segmentation errors can be classified as False Positives (FP), False Negatives (FN), over-segmentation, under-segmentation, and wrong partitioning of touching cells [43].

To handle such segmentation errors in tracking by detection methods, two strategies exist: 1) Generating overlapping segmentation masks and selecting the final set of segmentation masks in the tracking step [31, 32, 34, 36, 42, 44]. 2) Using non overlapping segmentation masks and detecting and correcting segmentation errors [24, 28, 30, 33, 35, 39, 40, 45–47]. The first strategy is computationally expensive as several segmentation hypothesis are competing. For the second strategy, semi-automated methods with manual data curation [45, 46, 48] and automated methods [24, 28, 30, 33, 35, 39, 40, 47] have been proposed.

While semi-automated methods need manual effort for error correction, automated segmentation correction approaches often require a learning step. For instance, classifiers that estimate the number of objects per segmentation mask are trained [30, 33], where no objects correspond to FPs, and more than one object to an under-segmentation error. Another approach is to train a support vector machine to distinguish mitosis from over-segmentation [39]. While there are approaches resolving multiple over- or under-segmented cells, they assume no mitosis events are occurring [40, 47]. To detect FPs, prior knowledge on the length of mitosis cycle [35], or on the expected track length [24] is used. Also, a two stage tracking is proposed to first construct short tracks and then resolve segmentation errors in the second step to yield the final tracks [28]. Besides that, uncertainty information is propagated through the segmentation and tracking pipeline to improve results [49].

In summary, current tracking approaches are either simple to apply but need manual error correction, or are able to correct segmentation errors but have a vast set of parameters to tune or need additional training data. In this work, we propose a compromise between the two sides, a simple to apply tracking algorithm, which needs no training and extensive parameter tuning, yet able to correct certain types of segmentation errors.

The main contributions are: a) We propose a tracking approach able to handle the segmentation errors under- and over-segmentation with more than two objects involved and FN. b) We make our Python code available as open source https://git.scc.kit.edu/KIT-Sch-GE/ 2021-cell-tracking. c) We compare our tracking approach to three other tracking approaches which performed competitive on the Cell Tracking Challenge (CTC) http:// celltrackingchallenge.net/ [5, 50] and investigate how robust the selected tracking approaches perform, when the segmentation quality decreases. d) We show that our tracking algorithm performs well on a vast set of 2D and 3D data sets of the CTC using the same parametrization of our manually tunable parameters for all data sets.

## Materials and methods

In this section, we describe our tracking algorithm, which is able to process 2D and 3D image sequences, in more detail. To create a tracking by detection algorithm, our proposed tracking algorithm can be combined with an arbitrary segmentation algorithm, which predicts instance segmentation masks. Moreover, the tracking can be included in a full image analysis pipeline which typically consists of sample preparation and imaging, cell segmentation, cell tracking, and subsequent analysis [51, 52].

Our tracking algorithm is based on the following assumptions: The cell movement is small compared to the overall image size and the majority of segmentation masks are segmenting single cells correctly. The cell movement assumption is motivated by the need of a reasonable temporal resolution of the image sequence for a detailed analysis of cell lineage or cell behavior. The segmentation assumption is motivated by the availability of reasonably well-performing segmentation approaches [7–11]. Due to potentially occurring segmentation errors, we refer to segmentation masks as segmented objects and not as cells, as the segmentation masks can contain an as cell detected artifact, only parts of a cell, a single cell or several cells.

We split the task of cell tracking into three steps: tracklet step, matching step, and post-processing step. In the tracklet step, the segmented objects are coarsely followed over time to find potential objects belonging to the same track. In the matching step, the segmented objects are assigned to tracks by solving a graph-based optimization problem. The graph models cell behavior including appearance, disappearance, movement, and mitosis as well as the segmentation errors over- and under-segmentation and FN. Lastly, a post-processing step is applied to correct segmentation errors. An overview of the tracking pipeline based on an example is shown in Fig 1.

### Step 1: Tracklet step

Based on the cell movement assumption, segmented objects belonging to the same track should be spatially close between successive time points. Similar as in [9], we define for each segmented object a rectangular shaped region of interest (ROI), which size is derived from the average size of the segmentation masks, to find objects which could belong to the same track at successive time points. The ROI is propagated over time by estimating a displacement between successive frames using a phase correlation [53]. We consider segmented objects which overlap with the propagated ROI as matching candidates which will be linked in the matching step.

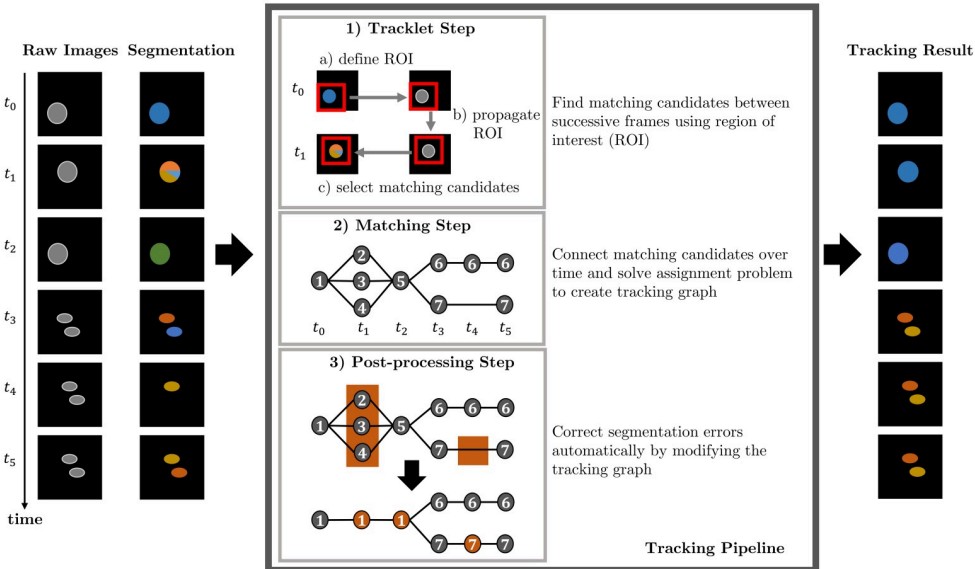

**Fig 1. Tracking pipeline.** Steps of our proposed tracking algorithm based on an input image sequence with erroneous segmentation data. After processing the image sequence through the tracking pipeline, the cells are tracked and segmentation errors are corrected. The node IDs in the tracking graph indicate the assigned track ID to the segmented objects.

## Step 2: Matching step

We model segmented objects and their matching candidates as nodes in a directed graph. The graph models the cell behavior: appearance, disappearance, movement, and mitosis as well as the segmentation errors: FNs, under- and over-segmentation. By finding optimal paths through this graph, the segmented objects are linked over time.

Let $\mathcal{G} = (\mathcal{V}, \mathcal{E})$ be a directed graph with a set of nodes $\mathcal{V} = \{u, v, w, \ldots\}$ and a set of edges $\mathcal{E} = \{(u, v)\}$ connecting pairs of nodes $u$ and $v$. Edges $(u, v)$ are directed, starting from node $u$ and ending in node $v$.

**Nodes.** We define node types to model cell behavior and segmentation errors:

- $q^-$: source node

- $q^+$: sink node

- $o_{\cdot,\cdot}$: object nodes modeling segmented objects

- $s_{\cdot,\cdot}$: split nodes modeling mitosis / over-segmentation errors

- $m_{\cdot,\cdot}$: merge nodes modeling under-segmentation errors

- $x_{\cdot,\cdot}$: skip nodes modeling FNs

- $d_{\cdot}$: delete nodes modeling disappearing objects

- $a_{\cdot}$: appear nodes modeling appearing objects

A specific node in the graph is referred to as $v_{i,t}$, where $v$ is the node type and $i$ is an unique identifier referencing a segmented object and $t$ a time point.

For each segmented object $i$ at time point $t$ a corresponding object node $o_{i,t}$ is added to the graph. To link tracks with missing segmentation masks over a maximum time span of $\Delta t$, we add skip nodes for each segmented object from time point $t$ at $\Delta t - 1$ successive time points. For each time point $t$ an appearance node $a_t$ is added to model appearing objects at time point $t + 1$, whereas a delete node $d_t$ is added for each time point $t$ to model disappearing objects at time point $t - 1$. Mitosis and over-segmentation errors are modeled by adding for each object node and skip node at time point $t$ a split node $s_{i,t+1}$ at time point $t + 1$. Under-segmentation errors are modeled by adding a merge node $m_{i,t-1}$ at time point $t-1$ for each object node and each skip node at time point $t$. The source node $q^-$ is added before the first time point and a sink node $q^+$ is added after the last time point of the considered set of time points $\mathcal{T}$.

**Edges.** The nodes are connected by directed edges to model events, such as linking segmented objects between successive time points. We allow directed edges between the following node types, where $u$: $\{v, w\}$ means edges starting from node type $u$ can end in the node types $v$ and $w$:

- $q^-$: $\{a_., o_{.,.}\}$

- $q^+$: $\{\}$

- $o_{.,.}$: $\{d_., m_{.,.}, o_{.,.}, s_{.,.}, q^+, x_{.,.}\}$

- $s_{.,.}$: $\{o_{.,.}\}$

- $m_{.,.}$: $\{d_., o_{.,.}\}$

- $x_{.,.}$: $\{m_{.,.}, o_{.,.}, s_{.,.}, q^+, x_{.,.}\}$

- $d_.$: $\{q^+\}$

- $a_.$: $\{d_., o_{.,.}, s_{.,.}\}$

Fig 2 shows the constructed graph based on the image sequence with erroneous segmentation from Fig 1.

Connecting all object nodes and skip nodes at time point $t$ naïvely to all other object nodes at time point $t + 1$, would result in a quadratically growing number of edges. To reduce the number of edges in the graph, we use the matching candidates from the tracklet step and connect nodes only to the nodes corresponding with its matching candidates. This is applied to the split and merge nodes as well, by connecting a split node $s_{i,t+1}$ only to the object nodes at $t + 1$, the object node $o_{i,t}$ or skip node $x_{i,t}$ is connected to. A merge node $m_{i,t}$ is only connected to the object nodes and skip nodes at $t$, the object node $o_{i,t+1}$ is connected to. A visualization how nodes are connected is shown in Fig 2. The used costs functions are introduced in more detail in the following.

**Formulation as coupled minimum cost flow problem.** In theory the graph could be spanned over the full time span of an image sequence, however, for data sets with many cells and time points this would lead to large optimization problems which need to be solved. Therefore, we solve smaller optimization problems by dividing the image sequence in smaller time spans and constructing graphs which overlap in time.

In the following, optimal paths through the graph are found by solving a coupled minimum cost flow problem. Our formulation is most similar to the coupled minimum cost flow problem [29], which we extend such that many to one and one to many links are possible as well as introducing skip nodes. Therefore, over- and under-segmentation of two or more objects as well as missing segmentation masks are modeled in the graph. To find optimal paths through the graph, a flow variable $z^f(u, v)$ is defined for each edge $(u, v)$, where $z^f(u, v) \in \mathbb{N}_0$. The

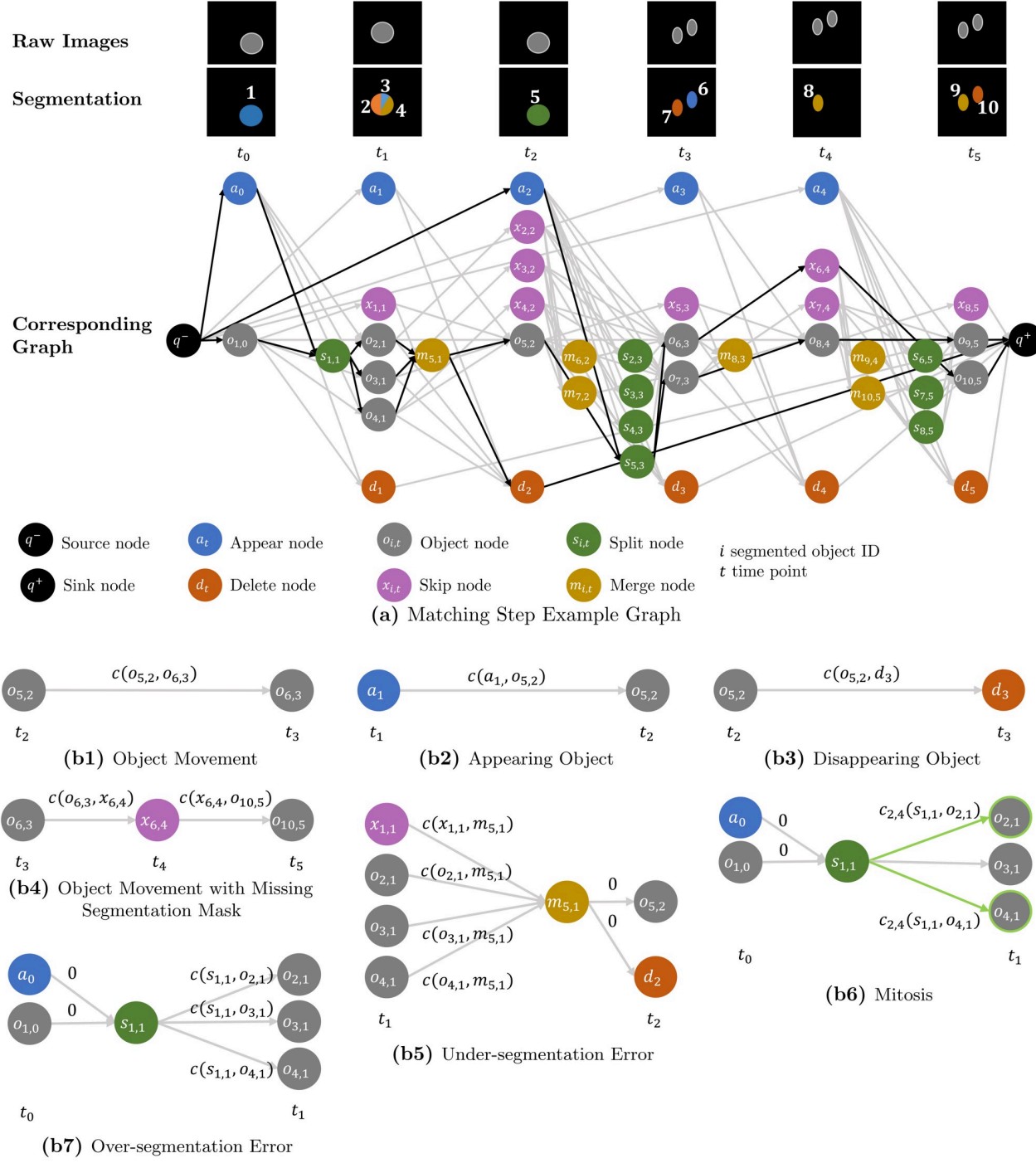

**(a)** Matching Step Example Graph

**(b1)** Object Movement

**(b2)** Appearing Object

**(b3)** Disappearing Object

**(b4)** Object Movement with Missing Segmentation Mask

**(b5)** Under-segmentation Error

**(b6)** Mitosis

**(b7)** Over-segmentation Error

**(b)** Modeling Cell Behavior and Segmentation Errors

**Fig 2. Matching step.** (a) shows a graph constructed from an image sequence with erroneous segmentation. Each segmented object is assigned an unique ID $i$. Nodes corresponding to a segmented object share the same ID $i$, however, depending on the node type these nodes are assigned to different time points $t$ in the graph. We link segmented objects over a maximum time span of $\Delta t = 2$ frames by adding for each object node $o_{i,t}$ a skip node $x_{i,t+1}$, which models a missing segmentation mask. The segmented objects are assigned to tracks by finding optimal paths—highlighted in black —through the graph. (b) visualizes how cell behavior and segmentation errors are modeled in the graph example (a). Annotations $c(\cdot, \cdot)$ on the edges are assigned edge costs. To model mitosis, edges which are connected to pairs of "daughter" nodes are pairwise coupled—highlighted in green.

optimization problem is given as

$$\min_{z^f(u,v)} \sum_{(u,v)\in\mathcal{E}} c(u,v) z^f(u,v)$$

subject to :

$$g_{i'}(z^f(u,v)) = 0, \ i' = 0,\ldots,N$$

$$h_{j'}(z^f(u,v)) \le 0, \ j' = 0,\ldots,M$$

(1)

where $c(u,v)$ is a cost and $g_{i'}$ are equality constraints and $h_{j'}$ inequality constraints, which will be introduced in the following.

A flow conservation constraint is added for all nodes apart from source node and sink node:

$$\sum_{u\in\mathcal{V}} z^f(u,v) = \sum_{w\in\mathcal{V}} z^f(v,w).$$

(2)

Flow requirements enforce a flow of a fixed number of units through the graph. We ensure that each segmented object is assigned to a track, by enforcing a flow of one trough each object node $o_{i,t}$ and setting the flow from the source node $q^-$ to the total number of segmented object nodes

$$\sum_{u\in\mathcal{V}} z^f(q^-,u) = \sum_{t'\in\mathcal{T}} |\mathcal{O}_{t'}|,$$

$$\sum_{u\in\mathcal{V}} z^f(u,o_{i,t}) = 1,$$

$$z^f(q^-,a_t) = |\mathcal{O}_{t+1}|,$$

(3)

where $\mathcal{T}$ is the set of all time points in the graph and $|\mathcal{O}_t|$ is the number of object nodes at time point $t$.

The flow over an edge $(u,v)$ is restricted by a maximum capacity constraint $b(u,v)$:

$$0 \le z^f(u,v) \le b(u,v).$$

(4)

Edges connected to at least one skip $x_{i,t}$ or object node $o_{i,t}$ have a capacity of one

$$b(u,o_{i,t}) = b(u,x_{i,t}) = b(o_{i,t},v) = b(x_{i,t},v) = 1.$$

(5)

To model over- and under-segmentation of more than two objects, the capacity of edges connecting merge nodes $m_{i,t-1}$ to delete nodes $d_t$ and appear nodes $a_t$ to split nodes $s_{i,t+1}$ depends on the number of edges ending in the merge node and edges starting from the split node, respectively:

$$b(m_{i,t-1},d_t) = |\{(v_{.,t-1},m_{i,t-1}) \mid v_{.,t-1} \text{ connected to } m_{i,t-1}\}|,$$

$$b(a_t,s_{i,t+1}) = |\{(s_{i,t+1},v_{.,t+1}) \mid v_{.,t+1} \text{ connected to } s_{i,t+1}\}|.$$

(6)

The capacity of edges connecting the source node $q^-$ to appearance nodes $a_t$ depends on the number of segmented objects at time point $t+1$, whereas the capacity of edges connecting delete nodes $d_t$ to the sink node $q^+$ depend on the number of segmented objects at time

points $\{t - \Delta t, \ldots, t\}$

$$
\begin{aligned}
b(q^-, a_t) &= |\mathcal{O}_{t+1}|, \\
b(a_{t-1}, d_t) &= |\mathcal{O}_t|, \\
b(d_t, q^+) &= \sum_{t'=t-\Delta t}^{t} |\mathcal{O}_{t'}|,
\end{aligned}
\tag{7}
$$

where $|\mathcal{O}_t|$ is the number of object nodes at time point $t$. The sum of the capacity constraint $b(d_t, q^+)$ results from the added skip nodes which enable linking segmented objects over a maximum time span $\Delta t$. For $\Delta t = 1$, no skip nodes are added resulting in $b(d_t, q^+) = |\mathcal{O}_t| + |\mathcal{O}_{t-1}|$, providing a large upper bound. For $\Delta t = 2$ for each object node a skip node is added, allowing a flow from an object node at $t - 2$ over its skip node to a merge node at $t - 1$, which is connected to the delete node $d_t$. To provide a large enough upper bound, the number of object nodes from time point $t - 2$ is added to the maximum capacity:

$$
b(d_t, q^+) = \sum_{t'=t-2}^{t} |\mathcal{O}_{t'}| = |\mathcal{O}_t| + |\mathcal{O}_{t-1}| + |\mathcal{O}_{t-2}|.
$$

To model under-segmentation of two or more objects, for each merge node $m_{i,t-1}$ the following constraints are added:

$$
\begin{aligned}
z^f(m_{i,t-1}, o_{i,t}) - z^f(m_{i,t-1}, d_t) &\leq 0, \\
z^f(v_{\cdot,t-1}, m_{i,t-1}) - z^f(m_{i,t-1}, o_{i,t}) &\leq 0 \ \forall \ v_{\cdot,t-1} \ \text{connected to} \ m_{i,t}.
\end{aligned}
\tag{8}
$$

Combining Eqs 4 and 8, we derive

$$
0 \leq z^f(v_{\cdot,t-1}, m_{i,t-1}) \leq z^f(m_{i,t-1}, o_{i,t}) \leq z^f(m_{i,t-1}, d_t).
$$

For a flow $z^f(v_{\cdot,t-1}, m_{i,t-1})$ from a node $v_{\cdot,t-1}$ to the merge node $m_{i,t-1}$ larger than zero, the flow from the merge node to the object node $z^f(m_{i,t-1}, o_{i,t})$ and the flow from the merge node to the delete node $z^f(m_{i,t-1}, d_t)$ need to be at least as large. The flow conservation constraint Eq 2 enforces the same flow into a node and from a node, resulting in a flow of at least two through the merge node $m_{i,t-1}$ or zero.

To model over-segmentation into two or more objects, for each split node the following constraints are added:

$$
\begin{aligned}
-z^f(a_t, s_{i,t+1}) + z^f(o_{i,t}, s_{i,t+1}) &\leq 0, \\
-z^f(o_{i,t}, s_{i,t+1}) + z^f(s_{i,t+1}, v_{\cdot,t+1}) &\leq 0 \ \forall \ v_{\cdot,t+1} \ \text{connected to} \ s_{i,t+1}.
\end{aligned}
\tag{9}
$$

Similar to before, we derive by combining Eqs 4 and 9

$$
0 \leq z^f(s_{i,t+1}, v_{\cdot,t+1}) \leq z^f(o_{i,t}, s_{i,t+1}) \leq z^f(a_t, s_{i,t+1}).
$$

For a flow $z^f(s_{i,t+1}, v_{\cdot,t+1})$ from the split node $s_{i,t+1}$ to a node $v_{\cdot,t+1}$ larger than zero, the flow from the object node to the split node $z^f(o_{i,t}, s_{i,t+1})$ and the flow from the appear node to the split node $z^f(a_t, s_{i,t+1})$ need to be at least as large. The flow conservation constraint Eq 2 enforces the same flow into a node and from a node, resulting in a flow of at least two through the split node $s_{i,t+1}$ or zero.

To distinguish an over-segmentation from a mitosis and assign different cost functions, we construct all pairs of "daughter" nodes $o_{j,t+1}$ & $o_{l,t+1}$ the "mother" node $s_{i,t+1}$ is connected to

and add pairwise coupled flow variables. We refer to those pairwise coupled flow variables as $z_{jl}^f(\cdot, \cdot)$, where $jl$ refers to the indices of the pair of coupled daughter nodes. From each mother cell at most one pair of daughter cells can emerge, which is modeled by connecting the split node $s_{i,t+1}$ to at most two daughter nodes

$$z_{jl}^f(s_{i,t+1}, o_{j,t+1}) \quad = z_{jl}^f(s_{i,t+1}, o_{l,t+1}).$$

$$\sum_j \sum_{\substack{l \\ l \neq j}} z_{jl}^f(s_{i,t+1}, o_{j,t+1}) \quad \leq 2.$$

(10)

In addition, a split node $s_{i,t+1}$ can either model a mitosis or an over-segmentation. We enforce this by adding for all pairs of flow variables that correspond to edges starting from $s_{i,t+1}$ an inequality constraint:

$$z_{jl}^f(s_{i,t+1}, o_{j,t+1}) - z^f(s_{i,t+1}, o_{l,t+1}) \leq 1.$$

(11)

The number of daughter pairs grows quadratically with the number of potential daughter cells. To reduce the number of pairwise coupled flow variables, we prune the number of potential mitosis pairs to $N_{\max} = 10$ for each segmented object, based on the mitosis cost which is given in Eq 15.

**Cost functions.** Compared to other approaches, we choose costs $c(u, v)$ based on positional features only. We extract for each segmented object based on its segmentation mask the mask centroid $\mathbf{p}_{i,t}$, a bounding box, and a set of mask points. The bounding box is spanned by the top left and bottom right coordinates of the segmentation mask $i$ at time point $t$ and contains all points within the spanned rectangle, it will be referred to as $\mathcal{B}_{i,t}$. The set of mask points is derived by calculating a distance transformation on the segmentation mask and will be referred to as $\mathcal{Q}_{i,t}$, where a single point will be referred to as $\mathbf{q}_{i,t}$. A visualization of the extracted features is shown in Fig 3.

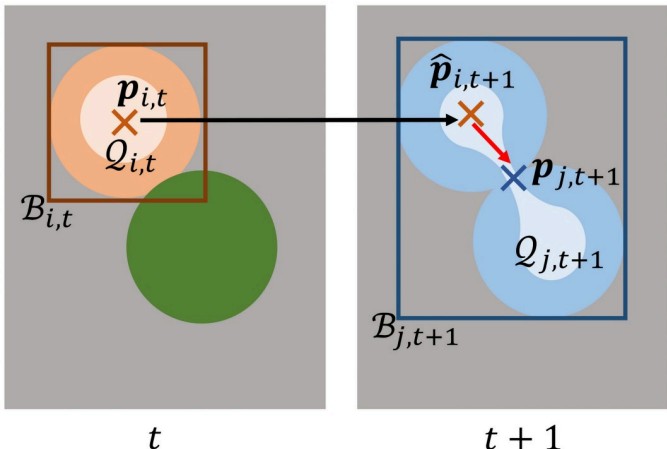

**Fig 3. Extracted features to link segmented objects.** Shown are two correctly segmented objects at time point $t$ and a single segmented object due to an under-segmentation error at time point $t + 1$. To calculate cost terms, for each segmentation mask $i$ at time point $t$ the mask centroid $\mathbf{p}_{i,t}$—shown as a cross –, a set of mask points $\mathcal{Q}_{i,t}$—shown in a lighter shade—and a bounding box $\mathcal{B}_{i,t}$—shown as a rectangle—are extracted. The Euclidean distance between the mask centroid $\mathbf{p}_{j,t+1}$ and the propagated mask centroid $\hat{\mathbf{p}}_{i,t+1}$ is large, which can result in wrong links. The minimal Euclidean distance between the propagated mask centroid $\hat{\mathbf{p}}_{i,t+1}$ and the set of mask points $\mathcal{Q}_{j,t+1}$, in contrast, is small.

The features are propagated over time by updating their position-based features with the estimated displacement $\mathbf{d}_{i,\cdot,\cdot}$ from the tracklet step:

$$
\begin{aligned}
\hat{\mathbf{p}}_{i,t+1} &= \mathbf{p}_{i,t} + \mathbf{d}_{i,t,t+1}, \\
\hat{\mathcal{B}}_{i,t+1} &= \{\mathbf{b}_{i,t} + \mathbf{d}_{i,t,t+1} \mid \mathbf{b}_{i,t} \in \mathcal{B}_{i,t}\}, \\
\hat{\mathcal{Q}}_{i,t+1} &= \{\mathbf{q}_{i,t} + \mathbf{d}_{i,t,t+1} \mid \mathbf{q}_{i,t} \in \mathcal{Q}_{i,t}\}.
\end{aligned}
\tag{12}
$$

Costs between object nodes model the movement of an object between successive time points:

$$
c(o_{i,t}, o_{j,t+1}) = \|\hat{\mathbf{p}}_{i,t+1} - \mathbf{p}_{j,t+1}\|_2,
\tag{13}
$$

where $\hat{\mathbf{p}}_{i,t+1}$ is the estimated mask centroid of object $i$ at time point $t + 1$ and $\mathbf{p}_{j,t+1}$ the mask centroid of object $j$ at time point $t + 1$. The edge costs involving skip nodes are defined as

$$
\begin{aligned}
c(o_{i,t}, x_{i,t+1}) &=
\begin{cases}
\|\mathbf{p}_{i,t} - \hat{\mathbf{p}}_{i,t+1}\|_2 = \|\mathbf{d}_{i,t,t+1}\|_2 & \text{if } \hat{\mathbf{p}}_{i,t+1} \notin \mathcal{B}_{j,t+1} \ \forall j \\
\theta & \text{else}
\end{cases}, \\[2mm]
c(x_{i,t+1}, x_{i,t+2}) &=
\begin{cases}
\|\hat{\mathbf{p}}_{i,t+1} - \hat{\mathbf{p}}_{i,t+2}\|_2 = \|\mathbf{d}_{i,t+1,t+2}\|_2 & \text{if } \hat{\mathbf{p}}_{i,t+2} \notin \mathcal{B}_{j,t+2} \ \forall j \\
\theta & \text{else}
\end{cases}, \\[2mm]
c(x_{i,t+1}, o_{j,t+2}) &= \|\hat{\mathbf{p}}_{i,t+2} - \mathbf{p}_{j,t+2}\|_2,
\end{aligned}
\tag{14}
$$

where $\theta$ is a large constant.

The mitosis costs for the pairwise coupled flow variables are defined as

$$
\begin{aligned}
c_1 &= \left\|\mathbf{p}_{i,t} - \frac{1}{2}(\mathbf{p}_{j,t+1} + \mathbf{p}_{l,t+1})\right\|_2, \\
c_2 &= \left|\|\mathbf{p}_{i,t} - \mathbf{p}_{j,t+1}\|_2 - \|\mathbf{p}_{i,t} - \mathbf{p}_{l,t+1}\|_2\right|, \\
c_3 &= \|\mathbf{p}_{j,t+1} - \mathbf{p}_{l,t+1}\|_2, \\
c_{jl}(s_{i,t+1}, o_{j,t+1}) &= c_{jl}(s_{i,t+1}, o_{l,t+1}) =
\begin{cases}
c_1 + c_2 & \text{if } c_3 \leq 1.5 b_{i,t} \\
\theta & \text{else}
\end{cases},
\end{aligned}
\tag{15}
$$

where $b_{i,t}$ is the length of the diagonal spanned by the top left and bottom right coordinate of the bounding box. The cost enforce that daughter cells have a similar distance to the mother cell, their average position is close to the previous position of the mother cell and the distance between the daughter cells is small. An estimated position of the mother cell is not used, as the displacement estimation which is based on appearance of image crops is unreliable, when one image crop shows a single mother cell and the other shows two daughter cells.

In case of over- or under-segmentation, costs based on mask centroids can lead to large cost terms, as the Euclidean distance between the propagated mask centroid of a correctly segmented object and the mask centroid of merged objects can be large, which is shown in Fig 3. To better link under- and over-segmented objects to their correctly segmented corresponding objects at successive time points, we use the set of mask points instead. For over-segmentation

we define the costs

$$c(s_{i,t}, o_{j,t+1}) = \begin{cases} \min(\{\|\hat{\mathbf{q}}_{i,t+1} - \mathbf{p}_{j,t+1}\|_2 \mid \hat{\mathbf{q}}_{i,t+1} \in \hat{\mathcal{Q}}_{i,t}\}) & \text{if } \mathbf{p}_{j,t+1} \in \hat{\mathcal{B}}_{i,t+1} \\ \theta & \text{else} \end{cases}, \qquad (16)$$

where $\hat{\mathcal{Q}}_{i,t}$ is the set of propagated mask points and $\hat{\mathbf{q}}_{i,t+1}$ a propagated mask point of the segmented object $i$ at time point $t$.

For under-segmentation, we define

$$c(o_{j,t}, m_{i,t}) = \begin{cases} \min(\{\|\mathbf{q}_{i,t+1} - \hat{\mathbf{p}}_{j,t+1}\|_2 \mid \mathbf{q}_{i,t+1} \in \mathcal{Q}_{i,t+1}\}) & \text{if } \hat{\mathbf{p}}_{j,t+1} \in \mathcal{B}_{i,t+1} \\ \theta & \text{else} \end{cases}, \qquad (17)$$

where $\hat{\mathbf{p}}_{j,t+1}$ is the predicted position of the segmented object $j$ at time point $t$ and $\mathcal{Q}_{i,t+1}$ the set of mask points of the segmented object $i$ at the next time point $t + 1$. Appearance costs depend on a threshold $\alpha$ and the minimum distance of the mask centroid $\mathbf{p}_{i,t}$ to the image border

$$c(a_{t-1}, o_{i,t}) = \min(\alpha, \min(\min(\mathbf{a} - \mathbf{p}_{i,t}), \min(\mathbf{p}_{i,t}))), \qquad (18)$$

disappear costs are defined similar

$$c(o_{i,t}, d_{t+1}) = \min(\alpha, \min(\min(\mathbf{a} - \mathbf{p}_{i,t}), \min(\mathbf{p}_{i,t}))), \qquad (19)$$

where $\mathbf{a}$ is the image size and $\min(\min(\mathbf{a} - \mathbf{p}_{i,t}), \min(\mathbf{p}_{i,t}))$ the minimal distance to the image border.

We set the parameter $\theta = 1000\alpha$, where $\alpha$ is derived from the largest edge of the default size of the ROI, which is provided in the section parameter selection. All other edges are assigned 0 cost. An overview of the calculated costs based on the graph example is shown in Fig 2.

To reduce the number of flow variables even further, edges with large costs are pruned. The formulated problem can be solved using integer linear programming with a standard optimization toolbox such as Gurobi [54].

The tracking graph is constructed by following the optimal paths through the graph and assigning segmented objects to tracks if their corresponding object nodes are connected by the same path. On nodes where several paths start/end, new tracks are created and the predecessor/successor information is kept.

## Step 3: Post-processing step

In the post-processing, over- and under-segmentation errors are resolved and missing segmentation masks are added to resolve FNs.

**Untangling problem.** After the matching step, tracks can be assigned to more than one predecessor and/or more than two successors as shown in Fig 1. These many to one and one to many assignments are now resolved, so each track has at most one predecessor and at most two successors to model mitosis. As the tracks are "untangled", we will refer to this step as untangling step. We transform the tracking graph by applying a set of modifications on the tracking graph which will be referred to as untangling operations: remove an edge, split a track, and merge tracks. The edge remove operation removes a single predecessor—successor link. The split operation splits a track into several tracks, whereas the merge tracks operation merges several tracks resulting in a single track. A visualization of the untangling operations is shown in Fig 4.

Different combinations of untangling operations lead to valid tracking graphs, which is shown in Fig 4. The problem is to select a combination of untangling operations, which we

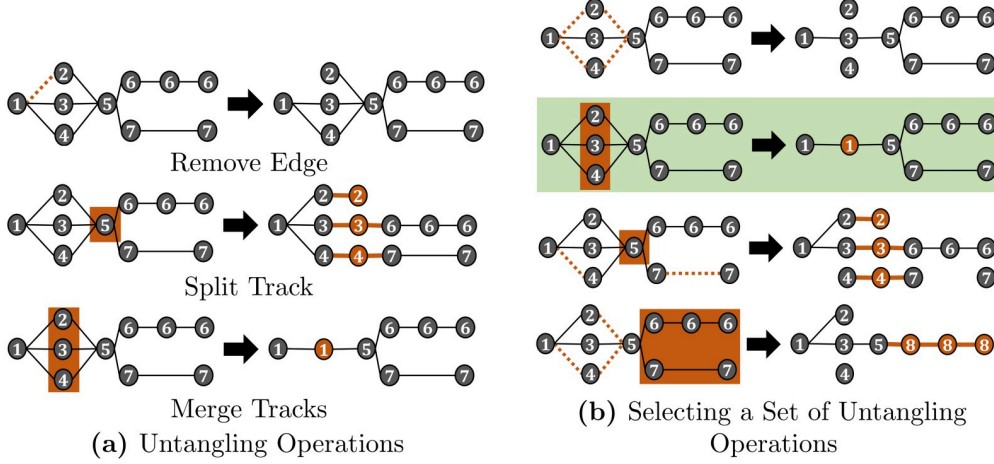

**Fig 4. Untangling problem.** The tracking graph is modified by applying untangling operations (a) such that each track has at most one predecessor and at most two successors—to model cell division. Different combinations of untangling operations, however, all lead to valid tracking graphs (b). We model the problem of selecting a set of untangling operations as an optimization problem and choose the set of untangling operations that induces the fewest modifications on the graph—highlighted in green.

model as an optimization problem

$$\min_z \sum_k c_k z_k \tag{20}$$

where $c_k$ are costs and $z_k$ variables referring to untangling operations on the graph.

The untangling operations $z_k$ are denoted as follows: an edge remove operation will be denoted as $z_{pn}^e$, where the predecessor track is $\omega_p$ and the successor track is $\omega_n$. Splitting a track $\omega_n$ into several tracks will be denoted as $z_n^s$, whereas merging a set of tracks will be denoted as $z_r^m$, where $r$ is a multi index that indicates a set of tracks.

To merge tracks, the tracks need to: a) share the same predecessors and successors, b) share the same successors and some tracks have no predecessor but begin after the track with a predecessor starts, or c) share the same predecessors and some tracks have no successors but end before the track with successors end. Based on the aforementioned conditions, we construct all possible sets of mergeable tracks.

Moreover, each track should have at maximum one predecessor and at maximum two successors. This is modeled by two constraints, one for the predecessor side and one for the successor side for each track. The number of predecessors of a track $\omega_n$ is referred to as $|\mathcal{P}_n|$ whereas the number of successors of a track is referred to as $|\mathcal{S}_n|$. For each set of tracks $r$ that can be merged, indicated by $z_r^m$, the number of tracks sharing the same set of predecessors as track $\omega_n$ is denoted as $P_{n,r}$ and the number of tracks sharing the same set of successors as track $\omega_n$ as $S_{n,r}$. Furthermore, for each predecessor track of track $\omega_n$ all sets of tracks the predecessor track can be merged with are computed, where the set $\mathcal{M}(\mathcal{P}_n)$ consists of all possible sets of mergeable tracks that contain predecessor tracks of track $\omega_n$. Analogously for each successor track of track $\omega_n$ all sets of tracks the successor track can be merged with are computed, where the set $\mathcal{M}(\mathcal{S}_n)$ consists of all possible sets of mergeable tracks that contain successor tracks of track $\omega_n$.

For each track we add one inequality constraint to enforce at most one predecessor and one inequality constraint to enforce at most two successors. As the modification of one track

influences also its predecessors and successors, the predecessor inequality also includes the untangling operations on the predecessor tracks, whereas the successor inequality includes the untangling operations on the successor tracks. Furthermore, as tracks can be linked to more than one predecessor and more than two successors, the tracks which share the same predecessors or successors need to be considered in the inequality constraints as well. The predecessor inequality constraint for track $\omega_n$ is given as:

$$
\overbrace{\sum_{r \in \mathcal{M}_r}(P_{n,r}-1)z_r^m}^{\text{merge tracks}} - \overbrace{\sum_{w \in \mathcal{W}_n}z_w^s}^{\text{split tracks}} - \overbrace{\sum_{w \in \mathcal{W}_n}\sum_{p \in \mathcal{P}_w}z_{pw}^e}^{\text{remove edges to predecessors}}
$$
$$
+ \underbrace{\sum_{p \in \mathcal{P}_n}z_p^s}_{\text{split predecessors}} + \underbrace{\sum_{q \in \mathcal{M}(\mathcal{P}_n)}\min(0, -S_{q,n}+1)z_q^m}_{\text{merge predecessors}} \leq -|\mathcal{P}_n| + \max(1, |\{\mathcal{S}_p \mid p \in \mathcal{P}_n\}|),
$$

(21)

where $r$ and $q$ are multi indices referring to sets of mergeable tracks and $w$ and $p$ are indices referring to a single track. The variables to be optimized are the merge track variables $z_r^m$ and $z_q^m$, the split track variables $z_w^s$ and $z_p^s$, and the edge remove variables $z_{pw}^e$, where $z_r^m$ denotes merging the set of tracks $r$ into a single track, $z_w^s$ denotes splitting the track $\omega_w$ into several tracks, and $z_{pw}^e$ denotes removing the predecessor-successor link between the predecessor track $\omega_p$ and the successor track $\omega_w$. The set $\mathcal{M}_n$ contains all sets of tracks that can be merged with the track $\omega_n$, $P_{n,r}$ is the number of tracks of the set of mergeable tracks $r$ that share the same predecessors as $\omega_n$, $\mathcal{W}_n$ is a set which contains all tracks, including $\omega_n$, that can be merged with track $\omega_n$. $|\mathcal{P}_n|$ is the number of predecessors of track $\omega_n$, whereas $\mathcal{P}_w$ is the set of predecessors of track $\omega_w$. $S_{q,n}$ is the number of tracks of the set of mergeable tracks $q$ that have track $\omega_n$ as a successor. The total number of successors of the predecessors of track $\omega_n$ is given by $|\{\mathcal{S}_p \mid p \in \mathcal{P}_n\}|$.

The successor inequality constraint is given as:

$$
\overbrace{\sum_{r \in \mathcal{M}_n}(S_{n,r}-1)z_r^m}^{\text{merge tracks}} - \overbrace{\sum_{w \in \mathcal{W}_n}z_w^s}^{\text{split tracks}} - \overbrace{\sum_{w \in \mathcal{W}_n}\sum_{v \in \mathcal{S}_w}z_{wv}^s}^{\text{remove edges to successors}}
$$
$$
+ \underbrace{\sum_{v \in \mathcal{S}_n}z_v^s}_{\text{split successors}} + \underbrace{\sum_{q \in \mathcal{M}(\mathcal{S}_n)}\min(0, -P_{q,n}+1)z_q^m}_{\text{merge successors}} \leq -|\mathcal{S}_n| + 2|\{\mathcal{P}_v \mid v \in \mathcal{S}_v\}| + 1,
$$

(22)

where $r$ and $q$ are multi indices referring to sets of mergeable tracks and $w$ and $v$ are indices referring to a single track. The variables to be optimized are the merge tracks variables $z_r^m$ and $z_q^m$, the split track variables $z_w^s$ and $z_v^s$, and the edge remove variables $z_{wv}^e$, where $z_r^m$ denotes merging the set of tracks $r$ into a single track, $z_w^s$ denotes splitting the track $\omega_w$ into several tracks, and $z_{wv}^e$ denotes removing the predecessor-successor link between the predecessor track $\omega_w$ and the successor track $\omega_v$. The set $\mathcal{M}_n$ contains all sets of tracks that can be merged with the track $\omega_n$, $S_{n,r}$ is the number of tracks of the set of mergeable tracks $r$ that share the same successors as track $\omega_n$, $P_{q,n}$ is number of tracks of the set of mergeable tracks $q$ that have track $\omega_n$ as a predecessor, and $\mathcal{W}_n$ is a set which contains all tracks, including $\omega_n$, that can be merged with track $\omega_n$. $|\mathcal{S}_n|$ is the number of successors of track $\omega_n$, whereas $\mathcal{S}_w$ is the set of successors of track $\omega_w$. $P_{q,n}$ is the number of tracks of the set of mergeable tracks $q$ that have track $\omega_n$ as a predecessor. The total number of predecessors of the successors of track $\omega_n$ is given by $|\{\mathcal{P}_v \mid v \in \mathcal{S}_v\}|$.

A track can be merged with at most one set of tracks $r$, which we model by adding for each track a constraint

$$\sum_{r \in \mathcal{M}_r} z_r^m \leq 1. \tag{23}$$

In addition, if a set of tracks is to be merged, their edge remove operations are coupled, such that for merged tracks either all edges are removed on the predecessor or successor side or none. To enforce this, we construct from each set of mergeable tracks $r$ all pairs of tracks which share a predecessor or successor and add two constraints

$$
\begin{aligned}
z_{pv}^e - z_{pn}^e &\leq -z_r^m + 1, \\
-z_{pv}^e + z_{pn}^e &\leq -z_r^m + 1,
\end{aligned}
\tag{24}
$$

where $r = \{n, v, \ldots\}$ and the tracks $\omega_n$ and $\omega_v$ share the predecessor track $\omega_p$. The merge tracks and edge remove variables are constraint to be binary variables, whereas the split variables are of integer type to provide the number of tracks a track will be split into.

**Predecessor and successor inequality constraints example.** We illustrate the setup of the proposed inequality constraints from Eqs 21 and 22 for the track with track ID 5 from the tracking graph shown in Fig 4. The track is connected to three predecessor tracks with the track IDs 2, 3, 4 and two successor tracks with the track IDs 6 and 7. As the track does not share its predecessors or successors with other tracks, there are no tracks the track can be merged with, therefore, the set containing all sets of mergeable tracks is $\mathcal{M}_5 = \{\}$ and $\mathcal{W}_5 = \{5\}$. The set of predecessor tracks is $\mathcal{P}_5 = \{2, 3, 4\}$ and $|\mathcal{P}_5| = 3$, whereas the set of successor tracks is $\mathcal{S}_5 = \{5, 6\}$ and $|\mathcal{S}_5| = 2$.

The set containing all possible sets of mergeable tracks that contain predecessor tracks is $\mathcal{M}(\mathcal{P}_5) = \{\{2,3\}, \{2,4\}, \{3,4\}, \{2,3,4\}\}$, whereas the set containing all possible sets of mergeable tracks that contain successor tracks is $\mathcal{M}(\mathcal{S}_5) = \{\{6,7\}\}$. The predecessors of track 5 have only one successor, which is track 5, resulting in $|\{\mathcal{S}_p \mid p \in \mathcal{P}_5\}| = 1$. The successor tracks of track 5 have only one predecessor, which is track 5, resulting in $|\{\mathcal{P}_v \mid v \in \mathcal{S}_5\}| = 1$.

By merging predecessor tracks or successor tracks into a single track, the number of predecessors or successors a track is connected to changes. The change in the number of predecessors or successors if sets of them are merged is represented by the terms $(P_{n,r} - 1)$ and $\min(0, -S_{q,n} + 1)$ from Eq 21, and $(S_{n,r} - 1)$ and $\min(0, -P_{q,n} + 1)$ in Eq 22. For example, by merging the tracks $\{2, 3, 4\}$ into a single track, which is modeled by $z_{\{2,3,4\}}^m$, two predecessor links of track 5 are removed, as now instead of three predecessor tracks only one predecessor track is connected to it. Therefore, $z_{\{2,3,4\}}^m$ is multiplied by a factor of 2.

After inserting the terms in the inequality constraints, we derive for the predecessor inequality constraint of track 5

$$
\overbrace{0}^{\text{merge tracks}} - \overbrace{z_5^s}^{\text{split tracks}} - \overbrace{(z_{2,5}^e + z_{3,5}^e + z_{4,5}^e)}^{\text{remove edges to predecessors}}
$$
$$
+\underbrace{z_2^s + z_3^s + z_4^s}_{\text{split predecessors}} \underbrace{-2z_{\{2,3,4\}}^m - z_{\{2,3\}}^m - z_{\{2,4\}}^m - z_{\{3,4\}}^m}_{\text{merge predecessors}} \leq -3 + 1 = -2,
\tag{25}
$$

and for the successor inequality constraint

$$
\overbrace{0}^{\text{merge tracks}} - \overbrace{z_5^s}^{\text{split tracks}} - \overbrace{(z_{5,6}^e + z_{5,7}^e)}^{\text{remove edges to successors}}
$$
$$
+ \underbrace{z_6^s + z_7^s}_{\text{split successors}} \quad \underbrace{-z_{\{6,7\}}^m}_{\text{merge successors}} \leq -2 + 2 + 1 = 1.
$$

(26)

The successor inequality constraint Eq 26 is fulfilled without applying untangling operations, as the right hand side of the inequality constraint is 1. However, untangling operations need to be applied so the predecessor inequality constraint Eq 21 holds, as the right hand side of the inequality constraint is −2. This makes sense, as track 5 has three predecessors and two successors and the aim of the untangling step is to transform the tracking graph such that each track has at most one predecessor and at most two successors.

**Untangling costs.**   The untangling costs can be chosen arbitrarily. Here we propose simple cost terms based on the temporal length and number of merged tracks:

$$
\begin{aligned}
c_{pn}^e &= \gamma, \\
c_r^m &= \Delta\omega_r(N_r - 1), \\
c_n^s &= \Delta\omega_n,
\end{aligned}
$$

(27)

where $c_{pn}^e$ is the cost of removing the edge between the tracks $\omega_p$ and $\omega_n$, $c_r^m$ is the cost of merging the set of tracks $r$, $c_n^s$ is the cost of splitting track $\omega_n$, $\gamma$ is a constant, $N_r$ is the number of merged tracks, $\Delta\omega_n$ is the temporal length of the track $\omega_n$ and $\Delta\omega_r$ is the temporal length of the track after merging, respectively. For the chosen cost functions, merging $K$ tracks or splitting a track in $K$ parts over the same time span, results in the same change of the value of the objective function. In theory, over- and under-segmentation errors can be resolved by only applying merging and splitting of tracks. However, there can be constellations where removing edges provides better tracking results, for instance due to a wrong link assigned in the matching step. To define a cut off when removing edges is more beneficial than modifying tracks, we set $\gamma$ to $2\lceil\Delta t_{0.3}\Delta N_{0.99}\rceil$, where $\Delta t_{0.3}$ is the 0.3 quantile of the track length and $\Delta N_{0.99}$ the 0.99 quantile of the number of predecessors/successor links per track.

The set of untangling operations is selected by solving an integer linear program using a standard optimization toolbox such as Gurobi [54]. After solving the optimization problem, the untangling operations are applied to the selected tracks. Tracks are split by computing for each mask $z_n^s$ seed points, where $z_n^s$ is the value of the split variable from the optimization problem. Based on the seed points, a nearest neighbor approach is applied to the mask and each mask pixel is assigned to the closest seed point, resulting in $z_n^s$ segmentation masks. To merge tracks, their segmentation masks are concatenated for each time point.

**FN correction.**   Finally, we correct FN errors by adding segmentation masks to tracks with missing segmentation masks. We place the last available segmentation mask, before a FN error occurs, at positions computed from a linear interpolation between the available segmentation masks. In image sequences with touching cells, adding masks can lead to conflicts, where an interpolated mask overlaps with another segmentation mask. We resolve those mask conflicts by assigning conflicting pixels to the segmentation mask with the closest centroid.

**Table 1. Statistics of cell data sets.** Information about the number of frames, tracks and cells of the CTC data sets.

| Data Set | Fluo-N2DH-SIM+ | | Fluo-N3DH-SIM+ | |
|---|---|---|---|---|
| Sequence | 01 | 02 | 01 | 02 |
| $N$ frames | 65 | 150 | 150 | 80 |
| $N$ tracks | 95 | 107 | 81 | 117 |
| avg. number of cells/frame | 40 | 22 | 19 | 43 |
| min number of cells/frame | 30 | 8 | 6 | 30 |
| max number of cells/frame | 47 | 54 | 43 | 55 |

## Data sets

We select the cell data sets Fluo-N2DH-SIM+ and Fluo-N3DH-SIM+ from the CTC [5, 50] for evaluation, as they are publicly available and provide a fully annotated ground truth, i.e. segmentation masks are given for all cells as well as the cell lineage. Both cell data sets show synthetically generated human leukemia cells, where Fluo-N2DH-SIM+ is a 2D data set and Fluo-N3DH-SIM+ is a 3D data set. Per data set two image sequences are available which will be referred to as sequence 01 and 02. Statistics of the chosen data sets are shown in Table 1.

**Simulation of segmentation errors.** We modify a fixed fraction of $n\%$ of the ground truth segmentation masks, to simulate data sets with an erroneous segmentation. We model the segmentation errors FNs, under- and over segmentation, and the combination of the aforementioned segmentation errors. FNs are simulated by removing segmentation masks randomly, the resulting data sets are referred to as "FN error". Over-segmentation is simulated by splitting segmentation masks randomly in two parts and is referred to as "over-segmentation error". Under-segmentation is simulated by selecting neighboring segmentation masks randomly and merging them to a single mask by applying a morphological closing operation. Data sets showing this error type will be referred to as "under-segmentation error". Furthermore, the error types are mixed by combining FN, under- and over-segmentation errors equally so in total $n\%$ of the segmentation masks are modified, which is referred to as "mixed error".

FN and over-segmentation errors are simulated by drawing uniformly from the set of segmentation masks until a fraction of $n\%$ of the ground truth masks is modified. Under-segmentation errors, in contrast, are sampled by constructing neighbor pairs of segmentation masks and assigning them a sampling weight proportional to their distance. As a result, cells with a smaller distance have a higher probability to be merged, which is also the case in real segmentation data. Segmentation masks are merged iteratively until a fraction of $n\%$ of the ground truth masks is merged, also allowing more than two cells to be merged.

For each cell data set and image sequence we modify $n = 1, 2, 5, 10, 20\%$ of the ground truth masks and generate $N = 5$ runs for each defined segmentation error which results in a total of 400 data sets for evaluation. A visualization of a raw image with corresponding ground truth and simulated segmentation errors is shown in Fig 5.

## Evaluation measure

We evaluate the segmentation and tracking performance using the SEG, TRA, and DET measure [5] from the CTC. The SEG measure is the Jaccard similarity index, which is the quotient of the intersection of segmentation and ground truth over the union of the two. In the TRA measure, graphs are constructed from the ground truth and the tracking data. Nodes in these graphs represent segmented objects, whereas edges represent links between the segmented

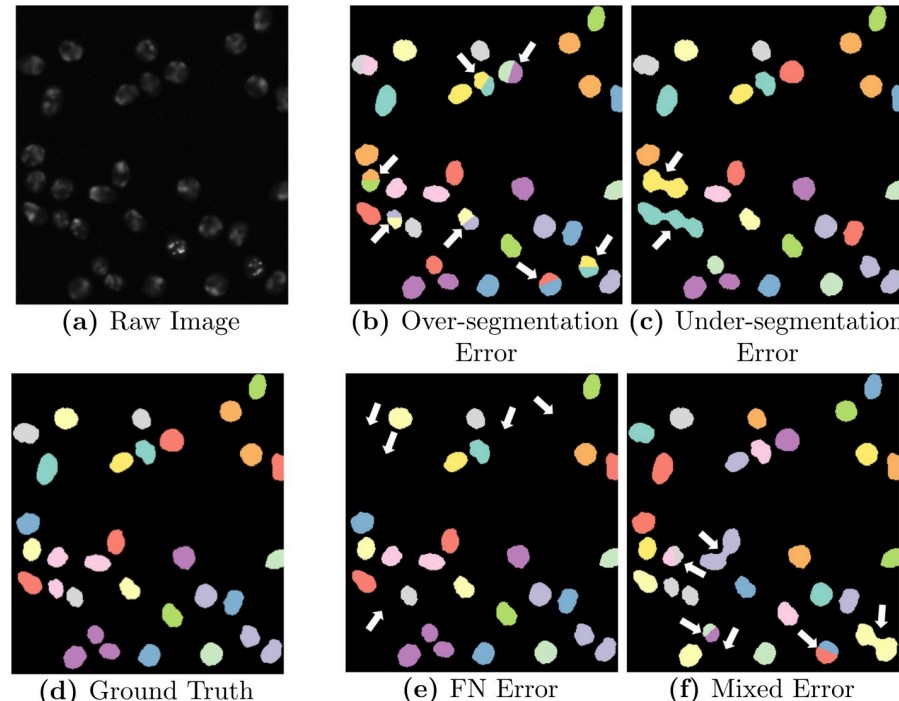

**Fig 5. Simulated segmentation errors.** Shown is a raw image of the Fluo-N2DH-SIM+ 01 data set with corresponding ground truth segmentation masks and modified segmentation masks with simulated segmentation errors, highlighted with white arrows.

objects over time. The tracking data graph is transformed into the ground truth graph by applying untangling operations: add/remove/split node, add/remove an edge and edit the edge semantic. Each graph operation results in a penalty, where adding nodes (FN) is penalized the most. The final measure is normalized between 0 and 1, where 1 means that ground truth graph and tracking data graph match perfectly. The DET measure is constructed similarly as the TRA measure, however, the penalties for modifying edges are set to zero.

## Compared tracking algorithms

We select three tracking by detection approaches from former CTC participants to compare our approach with: Mu-Lux-CZ, KIT-Sch-GE(1), and KTH-SE (http://celltrackingchallenge. net/participants/). All approaches provide an implementation which we used for comparison. The MU-Lux-CZ algorithm [15] is an overlap-based approach for 2D, which we extended to 3D. The tracking algorithm links segmentation masks between successive frames if their overlap is larger than a fixed threshold. Due to the simplicity of the algorithm, no automatic segmentation error correction is available. The KIT-Sch-GE(1) algorithm [9, 55] implements a coupled minimum cost flow algorithm which is capable to detect mitosis and handles FNs for short time spans. The KTH-SE algorithm [30, 56] uses the Viterbi algorithm to link cells. It includes segmentation error correction for FP, FN, over- and under-segmentation.

Compared to the tracking algorithm of Scherr et al. (Team KIT-Sch-GE(1) in CTC) [9, 55], our proposed tracking models FNs as skip nodes as well as over- and under-segmentation of two or more objects. Moreover, we propose different costs and the untangling post-processing step to correct segmentation errors automatically.

### Parameter selection

For the tracking approaches MU-Lux-CZ, KIT-Sch-GE(1), and KTH-SE we kept the same parameters as provided by their CTC submission, and only modified the algorithms such that they use the provided erroneous segmentation masks instead of using their own segmentation.

For our tracking algorithm we manually set two parameters: $\Delta t$, and the default ROI size. We set $\Delta t = 3$ and the default ROI size to twice of the average segmentation mask size. All other parameters are estimated automatically from the data or are based on these two parameters.

## Results

### Post-processing analysis

We investigate the influence of the post-processing steps, untangling tracks and FN correction, by modifying the post-processing step, while keeping all other steps the same. The FN correction step is replaced by creating short tracks without a predecessor for each track with missing masks, as the TRA measure yields for tracks with missing masks an error during TRA score computation. The untangling step is replaced by removing predecessor information of tracks with more than one predecessor and removing successor information of tracks with more than two successors. In the following, we will refer to the untangling step as untangle and the FN correction step as masks.

For under- and over-segmentation errors, tracking approaches without the untangling step, indicated by $\overline{\text{untangle}}$, perform worse as shown in Fig 6a and 6b for 2D and in Fig 7a and 7b for 3D data sets. On data sets with FN errors, tracking approaches without the FN correction step, indicated by $\overline{\text{masks}}$, perform worse which is shown for 2D data sets in Fig 6c and for 3D data sets in Fig 7c. When combining different segmentation error types, applying both post-processing steps performs best, which is shown in Figs 6d and 7d. Compared to segmentation only (No Tracking), the segmentation measures DET and SEG shown in Figs 6 and 7 increase after applying the tracking with the corresponding correction step in the post-processing.

### Tracking performance comparison

We compare the performance of our proposed approach including the proposed post-processing to the tracking approaches of KTH-SE, KIT-Sch-GE(1), and MU-Lux-CZ on erroneous segmentation data. The results are shown in Fig 8 for 2D data sets and in Fig 9 for 3D data sets.

For under-segmentation errors, our approach performs best, as shown in Figs 8b and 9b. On data sets with over-segmentation errors, our approach and the KTH-SE approach perform similarly, which is shown in Figs 8a and 9a. Both approaches lead to an increase in the segmentation measures SEG and DET on data sets with over-segmentation errors; applying the KTH-SE approach on any other type of erroneous segmentation leads to a decrease in the segmentation measures. In case of FN errors, our approach and the KIT-Sch-GE(1) approach perform similarly, as shown in Figs 8c and 9c. Also, both approaches yield higher scores in the segmentation measures DET and SEG compared to applying no tracking at all. For data sets with a combination of segmentation error types, our approach outperforms all other approaches, as shown in Figs 8d and 9d.

### Run-time comparison

We compare the run-time of the tracking algorithms when provided with perfect, ground truth segmentation data and when provided with erroneous segmentation data. As erroneous segmentation we choose the 20% mixed error segmentation data. We evaluated all tracking

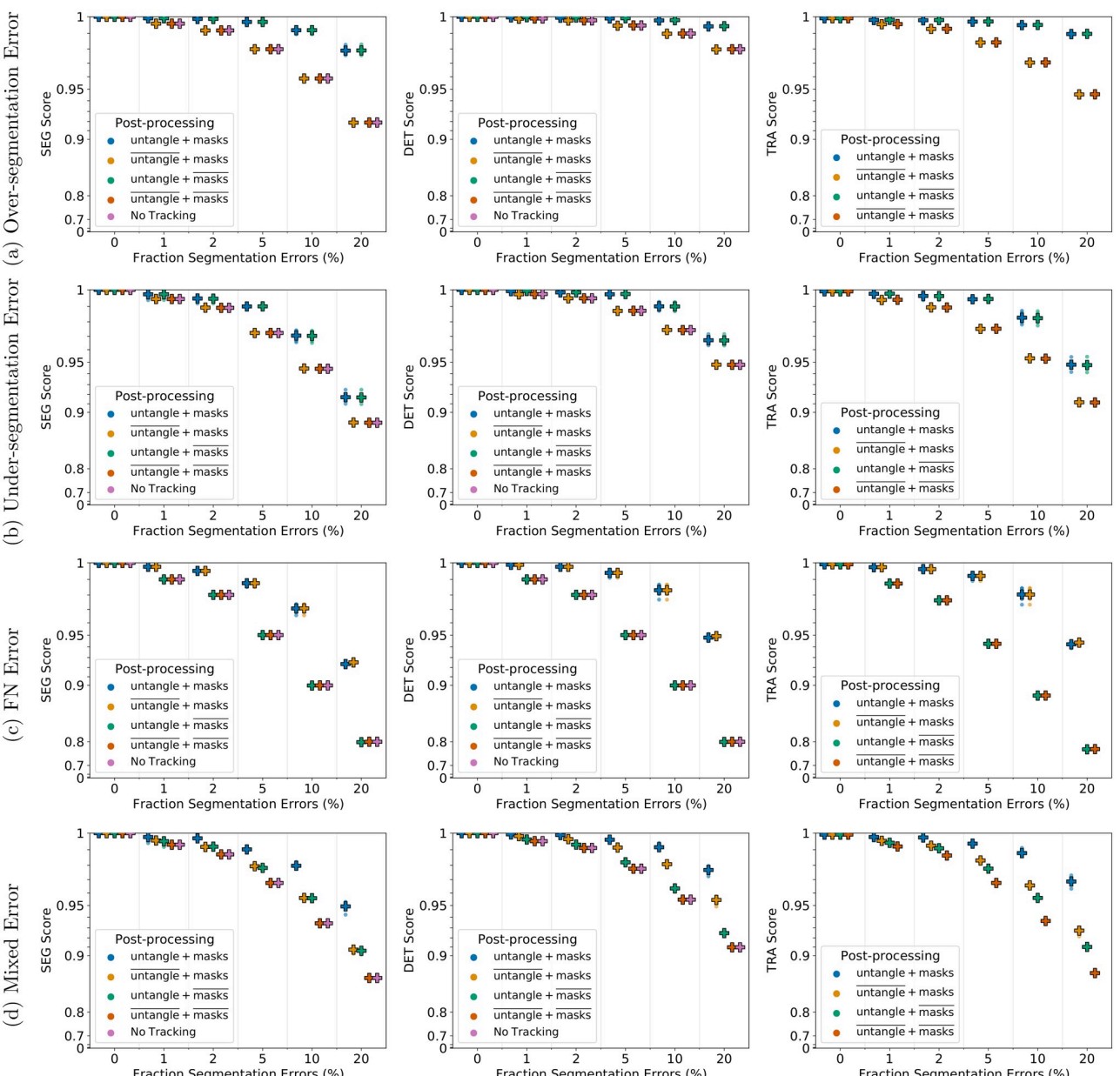

**Fig 6. Influence of the post-processing on Fluo-N2DH-SIM+ 01.** Scores of a single run are shown as circles, while + shows a CTC measure score averaged over *N* = 5 runs. Per run a fixed fraction of ground truth segmentation masks is modified randomly to simulate segmentation errors. "untangle" refers to the untangling step, which transforms the tracking graph such that each track has at most one predecessor and two successors, whereas "masks" refers to adding missing segmentation masks. Over lined post-processing steps $\overline{(\ldots)}$ indicate that the post-processing step is missing.

algorithms on a desktop computer with an Intel Core i7–6700 processor and 64GB of RAM running Python 3.7 and MATLAB 2018b in Windows 10.

For the KTH-SE algorithm, which is implemented in MATLAB, we used the `tic/toc` functionality of MATLAB for benchmarking, whereas for all other algorithms, which are implemented in Python, we used the `default_timer` of the `timeit` package. The run-times are shown in Table 2. The results show that the proposed method can track 2D and 3D data sets in reasonable times.

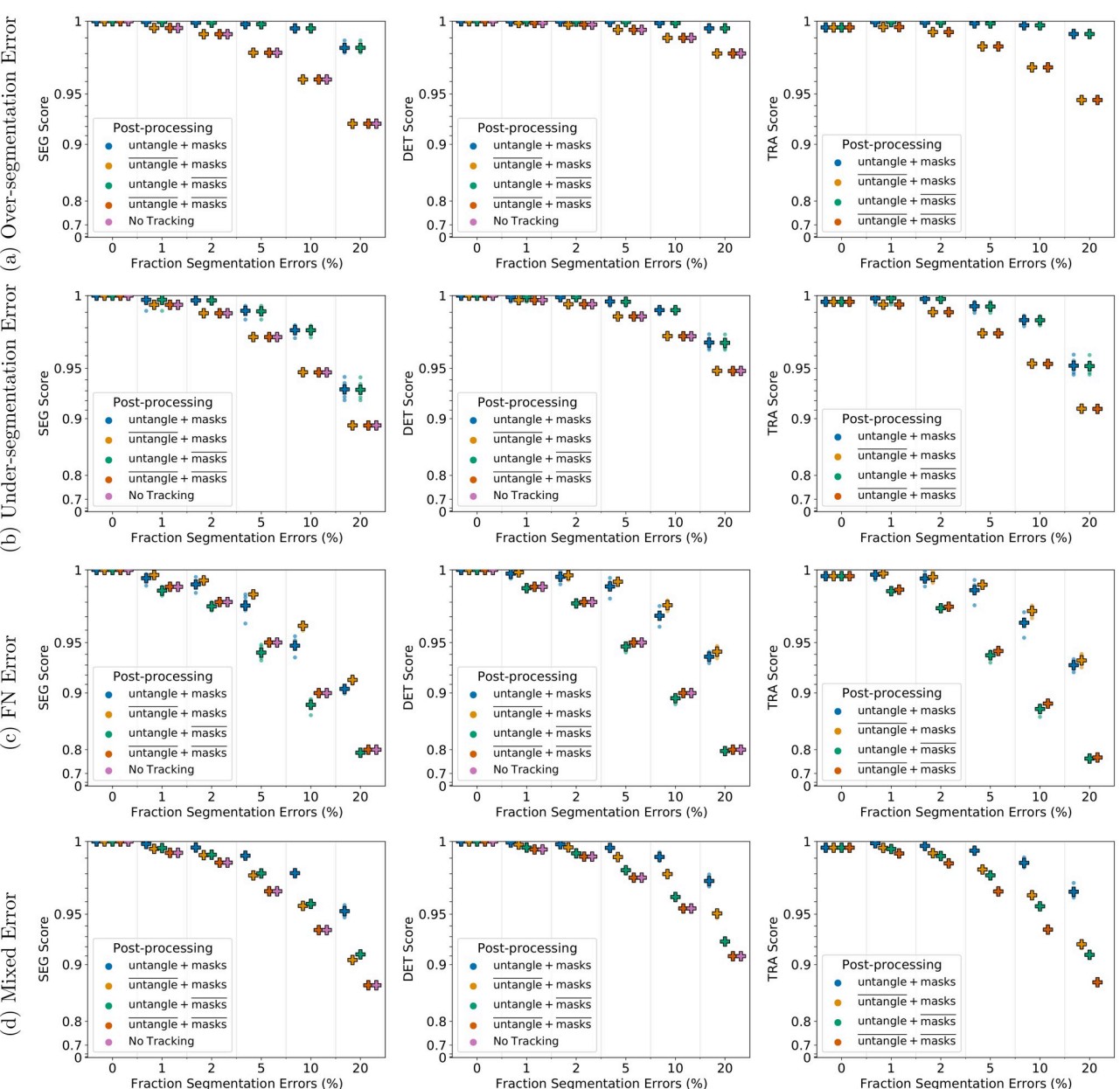

**Fig 7. Influence of the post-processing on Fluo-N3DH-SIM+ 01.** Scores of a single run are shown as circles, while + shows a CTC measure score averaged over $N = 5$ runs. Per run a fixed fraction of ground truth segmentation masks is modified randomly to simulate segmentation errors. "untangle" refers to the untangling step, which transforms the tracking graph such that each track has at most one predecessor and two successors, whereas "masks" refers to adding missing segmentation masks. Over lined post-processing steps $\overline{(\ldots)}$ indicate that the post-processing step is missing.

## Evaluation on the Cell Tracking Challenge

We evaluated the performance of our tracking algorithm on the 6[th] edition of the CTC. For segmentation, we chose a deep learning based segmentation approach which utilizes cell and neighbor distances [9, 57]. The derived segmentation masks and the raw image sequence were fed into our tracking algorithm. As described on the parameter selection section, we chose the same parametrization of the two manually tunable parameters for all data sets. The results of

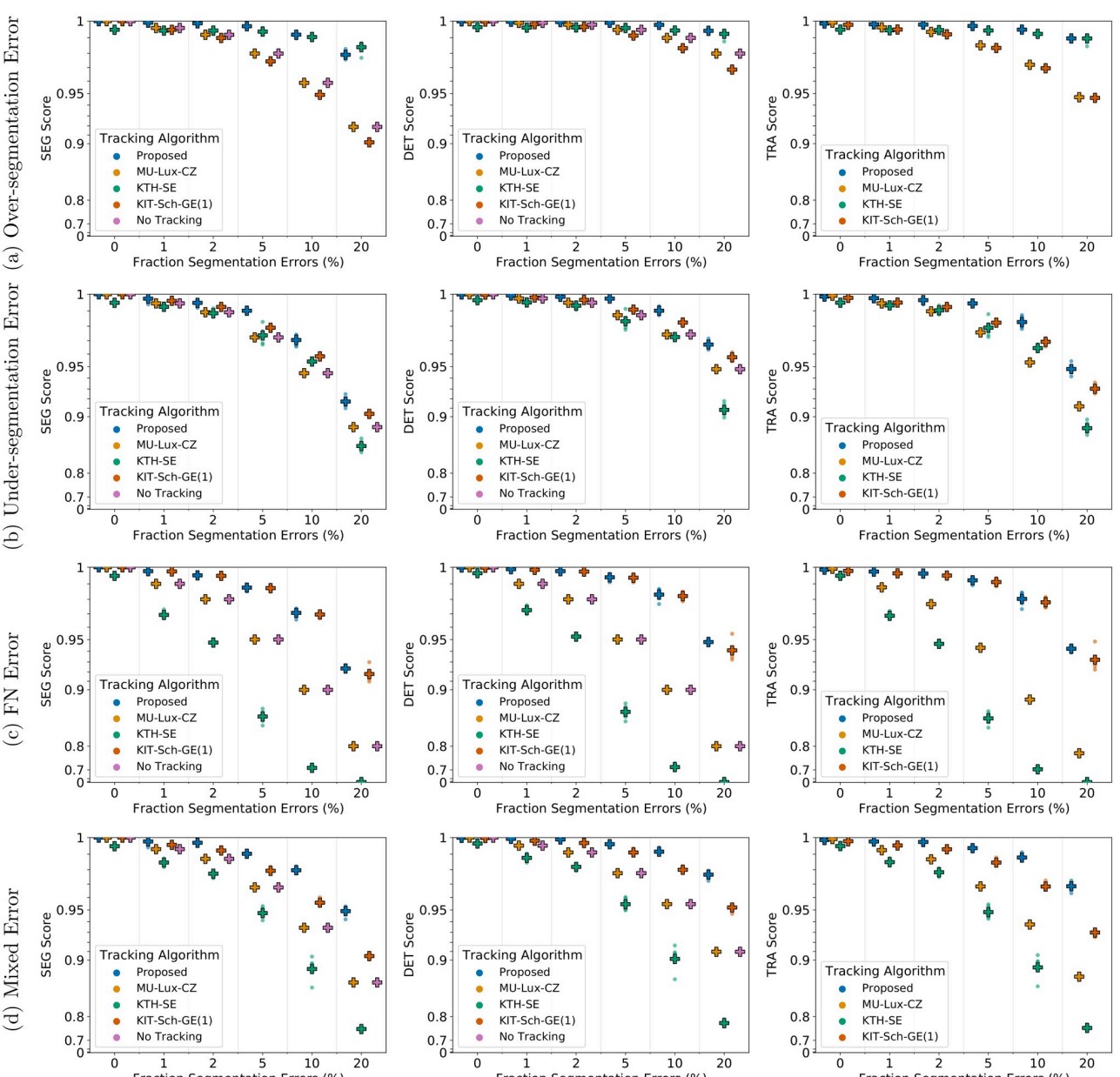

**Fig 8. Comparing tracking algorithms on Fluo-N2DH-SIM+ 01.** Shown are the CTC measure scores DET, SEG, and TRA of tracking algorithms on 2D data set Fluo-N2DH-SIM+ 01 when provided with the same erroneous segmentation data. Scores of a single run are shown as circles, while + shows a CTC measure score averaged over $N = 5$ runs.

the Cell Tracking Benchmark as team KIT-Sch-GE(2), with several top 3 ranks, are shown in Table 3.

## Discussion

Our proposed tracking method can correct the segmentation errors FN, under- and over-segmentation and yields a good tracking quality while having a reasonable run-time. The proposed post-processing with untangling step and FN correction improves in most cases the

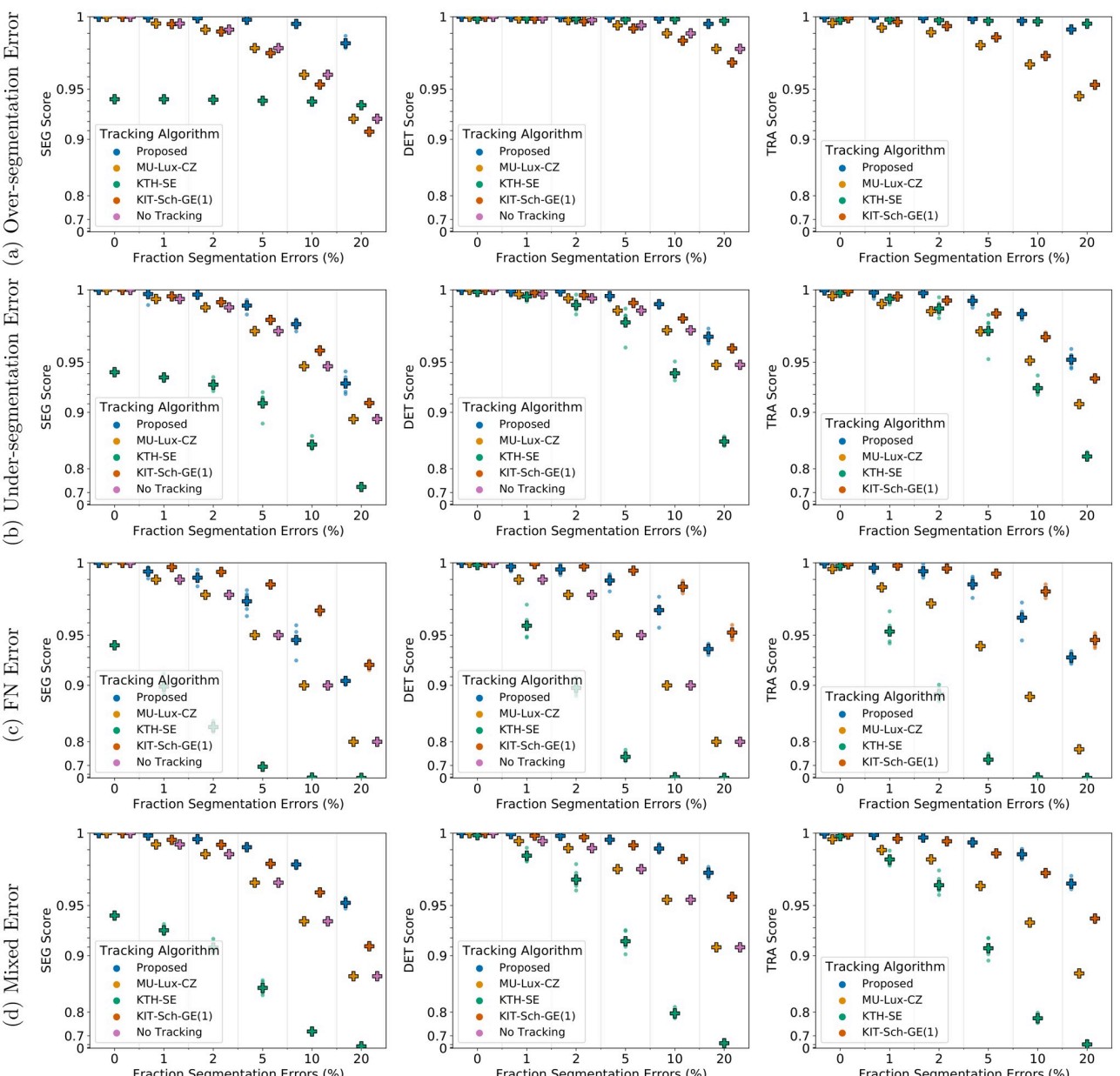

**Fig 9. Comparing tracking algorithms on Fluo-N3DH-SIM+ 01.** Shown are the CTC measure scores DET, SEG, and TRA of tracking algorithms on 3D data set Fluo-N2DH-SIM+ 01 when provided with the same erroneous segmentation data. Scores of a single run are shown as circles, while + shows a CTC measure score averaged over $N$ = 5 runs.

tracking and segmentation measure scores. However, when correcting FN errors on the 3D data set, shown in Fig 7c, we observe that applying both post-processing steps, referred to as untangle+ masks, performs worse compared to just applying the mask interpolation step, referred to as $\overline{\text{untangle}}$ + masks. We examined the TRA score in more detail and found that the scores of the untangle+ masks post-processing had more FPs. In some cases if a segmentation mask is missing and another segmentation mask is spatially close, the track with missing segmentation masks is linked to the spatially close track in the matching step, which is then

**Table 2. Run-times of tracking algorithms.** Run times of the tracking algorithms on 2D and 3D data sets when provided with perfect ground truth (GT) segmentation as well as when provided with erroneous segmentation data.

| Tracking Algorithm | Data Sets | | | |
|---|---|---|---|---|
| | Fluo-N2DH-SIM+ 01 | | Fluo-N3DH-SIM+ 01 | |
| | GT | Erroneous | GT | Erroneous |
| Proposed | 21.82 s | 22.94 s | 1238.38 s | 1179.69 s |
| MU-Lux-CZ | 25.79 s | 26.65 s | 1055.12 s | 1047.86 s |
| KTH-SE | 39.73 s | 44.22 s | 523.84 s | 404.34 s |
| KIT-Sch-GE(1) | 15.11 s | 19.56 s | 1004.63 s | 1585.43 s |

linked to two segmentation masks a few time points later when the object is segmented again. As a result the spatially close track has two predecessors assigned, which is resolved in the untangling step of the post-processing by splitting the track. In the TRA score this behavior is reported as FPs. An approach to resolve this, is using more complex cost functions in the untangling and the matching step, which for instance include information on the cell appearance.

All tracking approaches do not reach perfect measure scores of 1 on data sets with reduced segmentation quality. However, our approach is able to correct different types of segmentation errors, indicated by the increase of the segmentation scores DET and SEG compared to no tracking, without needing training data, a large set of parameters or extensive parameter tuning. Especially on data sets with a mixture of segmentation errors, our proposed method showed its potential as an "allrounder" method. To further improve tracking results, a manual correction step can be applied. The similar performance of our approach and the KIT-Sch-GE (1) approach, shown in Figs 8c and 9c, is due to the capability of both approaches to link tracks with missing masks over a maximum time span $\Delta t$, where both approaches set $\Delta t = 3$. Applying the MU-Lux-CZ tracking does not change the DET and SEG scores compared to applying no tracking, as this algorithm only links segmentation masks without any segmentation error correction. To our surprise the in the CTC [50] well-performing approach of KTH-SE drops in

**Table 3. Cell Tracking Benchmark (CTB) results (6<sup>th</sup> CTC edition).** Top 3 rankings as team KIT-Sch-GE(2) in the overall performance measure OP$_{CTB}$—average of SEG and TRA scores—are written in bold. The latest CTB leader board is available on the CTC website. State of the results: May 10<sup>th</sup> 2021.

| Data Set | SEG | TRA | Ranking TRA | OP$_{CTB}$ | Ranking OP$_{CTB}$ |
|---|---|---|---|---|---|
| BF-C2DL-HSC | **0.818** | **0.984** | **1** | **0.901** | **1** |
| BF-C2DL-MuSC | **0.777** | **0.967** | **3** | **0.872** | **1** |
| DIC-C2DH-HeLa | 0.778 | 0.918 | 7 | 0.848 | 8 |
| Fluo-C2DL-MSC | **0.617** | **0.749** | **4** | **0.683** | **3** |
| Fluo-C3DH-A549 | **0.849** | **1.000** | **1** | **0.925** | **1** |
| Fluo-C3DH-H157 | **0.878** | **0.980** | **2** | **0.929** | **2** |
| Fluo-C3DL-MDA231 | **0.710** | **0.884** | **1** | **0.797** | **1** |
| Fluo-N2DH-GOWT1 | 0.850 | 0.938 | 9 | 0.894 | 13 |
| Fluo-N2DL-HeLa | 0.883 | 0.993 | 1 | 0.938 | 10 |
| Fluo-N3DH-CE | 0.642 | 0.901 | 3 | 0.772 | 5 |
| Fluo-N3DH-CHO | 0.833 | 0.906 | 8 | 0.869 | 7 |
| PhC-C2DH-U373 | 0.876 | 0.975 | 9 | 0.925 | 10 |
| PhC-C2DL-PSC | **0.743** | **0.967** | **1** | **0.855** | **1** |
| Fluo-N2DH-SIM+ | 0.801 | 0.962 | 8 | 0.881 | 5 |
| Fluo-N3DH-SIM+ | **0.759** | **0.972** | **1** | **0.865** | **1** |

performance when provided with segmentation data which include under-segmentation and/ or FN errors. We examined the predicted tracking masks and observed that the approach removes some segmentation masks. Besides removing merged segmentation masks, sometimes also masks without added segmentation errors are removed by the tracking algorithm. In the TRA and DET measure, FN errors are penalized twice as much as not resolving an under-segmentation error. Hence, the MU-Lux-CZ approach, which applies no modification on the segmentation masks at all, performs better than the KTH-SE approach on all data sets which add FN errors and/or under-segmentation errors.

On the CTB, our tracking algorithm ranks several times within the top three without any manual parameter adaption. The difference in the ranking between $OP_{CTB}$ and TRA measure are due to the influence of the SEG measure—measuring how well ground truth mask and segmented masks align—on the $OP_{CTB}$. Improving the SEG score through the tracking—by splitting, merging, or adding segmentation masks—is only possible if the overall shapes of ground truth masks and segmented masks align well. Penalties in the SEG score by too large or too small segmentation masks can usually not be corrected by the tracking in a tracking by detection approach.

We would like to emphasize that the tracking performance depends on the instance segmentation which in turn depends on the image quality. To reduce the dependence on the image quality, the image quality can be improved substantially by applying image restoration methods before segmentation [58, 59]. In addition, instance segmentation approaches applicable to a broad range of imaging conditions exist [60]. While our tracking approach can be combined with an arbitrary instance segmentation approach, different instance segmentation approaches can be prone to different types and quantities of segmentation errors. Our results on simulated, erroneous segmentation data show, that our tracking algorithm can correct certain types of randomly occurring segmentation errors, however, with decreasing segmentation quality the tracking quality decreases as well.

In general, the tracking performance also depends on the temporal resolution of the image sequence. If the temporal resolution is high with respect to the cell movements—cell movements are small between frames with respect to the cell size—simple, nearest neighbor assignment is sufficient [61]. However, when the temporal resolution is restricted, e.g. to avoid photodamage, large cell movements between successive frames are possible. To assign the segmented cells correctly, more advanced approaches, such as graph-based approaches, are needed. As the results of the CTC show, our position-based costs perform well on a broad set of real world cell data sets, however, there are scenarios which will result in wrong assignments. For instance, consider two cells at time point $t$ which have swapped their positions at time $t+1$, which is impossible to detect using position-based costs. To resolve such cases, the tracking costs can be adapted using more complex features based on texture or morphology of single cells [25, 30, 37].

## Conclusion

We proposed a graph-based tracking approach with automatic correction of FN and under-/ over-segmentation errors of two or more segmented objects. Our approach neither needs training data to learn cost functions nor has a vast set of parameters that need manual tuning. We investigated the performance of our approach on a 2D and a 3D cell data set with synthetically degraded segmentation masks simulating FN, under-segmentation, over-segmentation and a combination of the aforementioned segmentation errors. We evaluated the tracking performance using the CTC measures DET, SEG, and TRA. For a fair comparison, we compared the performance of our tracking approach against three other tracking methods on the same

erroneous segmentation data. Our proposed tracking algorithm is capable to correct certain types of segmentation errors without requiring additional training steps or parameter tuning automatically. Furthermore, on data sets with under-segmentation or a combination of different segmentation errors, our approach outperformed all other approaches, especially a parameter heavy tracking algorithm with automated segmentation error correction.

Evaluated on a diverse set of 2D and 3D cell data sets from the CTC, our proposed tracking algorithm performed competitively without manual fine-tuning, showing its potential as a strong tracking baseline. Directions of future work could be incorporation of more complex cost functions for the matching and the untangling step, using more enhanced methods for the position estimation step, and application to other data sets. We envision that the steady improvement of automated cell tracking approaches concerning the accuracy, run time, and ease of applicability will lead to powerful tools to analyze cell behavior quantitatively. The derived insights on the cell behavior can then help to deepen our understanding of the mechanisms influencing cell migration or, for instance, how cell migration and the formation of structures depend on each other.

## Supporting information

**S1 Fig. Influence of the post-processing on Fluo-N2DH-SIM+ 02.**
(PDF)

**S2 Fig. Influence of the post-processing on Fluo-N3DH-SIM+ 02.**
(PDF)

**S3 Fig. Comparing tracking algorithms on Fluo-N2DH-SIM+ 02.**
(PDF)

**S4 Fig. Comparing tracking algorithms on Fluo-N3DH-SIM+ 02.**
(PDF)

**S1 Table. Run-times of tracking algorithms on image sequences 02.**
(PDF)

**S1 File. Data availability.**
(PDF)

## Author Contributions

**Conceptualization:** Katharina Löffler, Ralf Mikut.

**Data curation:** Katharina Löffler.

**Formal analysis:** Katharina Löffler.

**Funding acquisition:** Ralf Mikut.

**Investigation:** Katharina Löffler.

**Methodology:** Katharina Löffler.

**Project administration:** Ralf Mikut.

**Software:** Katharina Löffler, Tim Scherr.

**Supervision:** Ralf Mikut.

**Visualization:** Katharina Löffler.

**Writing – original draft:** Katharina Löffler.

**Writing – review & editing:** Katharina Löffler, Tim Scherr, Ralf Mikut.

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
