## [Decision Letter · Decision Letter 0]

20 Apr 2021

PONE-D-21-08192

A graph-based cell tracking algorithm with few manually tunable parameters and automated segmentation error correction

PLOS ONE

Dear Dr. Löffler,

Thank you for submitting your manuscript to PLOS ONE. After careful consideration, we feel that it has merit but does not fully meet PLOS ONE’s publication criteria as it currently stands. Therefore, we invite you to submit a revised version of the manuscript that addresses the points raised during the review process.

Please try to make the improvements according to the detailed suggestions of the three experts, which  you can find below.  Regarding  the additional literature mentioned by reviewers, the authors are, of course, free to decide, whether and which new references they will include in their manuscript.

We look forward to receiving your revised manuscript.

Kind regards,

Konradin Metze

Academic Editor

PLOS ONE

Journal Requirements:

Reviewers' comments:

Reviewer's Responses to Questions

**Comments to the Author**

1. Is the manuscript technically sound, and do the data support the conclusions?

Reviewer #1: Yes

Reviewer #2: Partly

Reviewer #3: Partly

2. Has the statistical analysis been performed appropriately and rigorously? 

Reviewer #1: N/A

Reviewer #2: Yes

Reviewer #3: No

3. Have the authors made all data underlying the findings in their manuscript fully available?

Reviewer #1: Yes

Reviewer #2: Yes

Reviewer #3: Yes

4. Is the manuscript presented in an intelligible fashion and written in standard English?

Reviewer #1: Yes

Reviewer #2: Yes

Reviewer #3: Yes

5. Review Comments to the Author

Reviewer #1: This paper presents an algorithm for automated cell segmentation and cell tracking, and evaluated the algorithm using data from the Cell Tracking Challenge. Overall, this is an interesting algorithm, combining graph based modeling and a couple of heuristics to produce an algorithm/pipeline for cell tracking. One great advantage of this algorithm is that it explicitly modeled several types of errors in the segmentations step (such as an object/cell disappearing, missing, dividing/mitosis, over-segmentation, under-segmentation), and showed that the subsequent operations on the graph and a post-processing step were able to correct these errors.

My main concerns and comments about this paper are as follows:

1. The algorithm sections are quite difficult to read. The notations are very complex. It feels like each set of equations were using very non-overlapping sets of symbols that represent different things. It is true that different sets of equations described different steps/componenets of the algorithm. If the notations can be significantly simplified, it will be a lot easier for readers to follow this paper and appreciate the algorithmic designs.

2. The figures are quite repetative in some sense, the same graph topology appearing in 4 different figures. Again, I understand that they convey different information on different steps of the algorithm, but I would be better to consolidate them into one figure with multiple sub-figures, so that it is easier for readers to understand the whole flow.

Reviewer #2: This is a good article and should be published provided the recommended modifications are fulfilled by the authors. The authors describe and compare their own segmentation algorithm with three established different segmentation algorithms and try to correct potentially incorrect results.

There are a few open questions / remarks, which should be briefly taken into account:

1. The algorithms seems to work on separated cell (cell lines only), and not on tissues (incompletely acquired objects). This should be discussed.

2. What kind of principle segmentation algorithm has been used (dynamic, static, Otsu, …) etc. or texture / transformation based methods (Fourier, Laplace, …)?

3. Are different cell types included in this study?

4. The authors should clearly distinguish the different mandatory compartments of the algorithms, namely (image acquisition (including image quality), object detection (differentiation potential Object / background), and object identification (object features) and classification (object class) for additional analysis (for example texture).

For example see and cite) (KAYSER, Klaus; BORKENFELD, Stephan; KAYSER, Gian. Digital Image Content and Context Information in Tissue-based Diagnosis. Diagnostic Pathology, [S.l.], v. 4, n. 1, dec. 2018. ISSN 2364-4893; GÖRTLER, Jürgen et al. Cognitive Algorithms and digitized Tissue – based Diagnosis. Diagnostic Pathology, [S.l.], v. 3, n. 1, july 2017. ISSN 2364-4893.

5. A few words should be addressed to future development (understanding) how structures 8cells) and functions (movements) might depend on each other. For example see and cite (Kayser, Klaus, Borkenfeld, Stephan, Fang, Wei-Kleiner, Kayser, Gian: Digital Pathology Where did – do – will you go? Content and Context Analysis of Communication in Digital Pathology. https://doi.org/10.17629/www.medical-journal-of-virtual-science.de-2021-1:2.

Reviewer #3: This paper presents a three-step cell tracking algorithm with segmentation error correction. The paper is well written, the content is clear, and the results seem to be promising. The tracking approach is compared with three other tracking approaches from the Cell Tracking Challenge CTC.

The drawback of the presentation is that the algorithm was only tested on two of 19 available datasets from the CTC. Another disadvantage is that only the tracking accuracies TRA are presented. With this minimal comparison, the evaluation of the proposed new algorithm cannot be performed satisfactorily.

Therefore, I would suggest adding further samples and adding the values for segmentation accuracies, detection accuracies, cell segmentation benchmarks and cell tracking benchmarks.

6. PLOS authors have the option to publish the peer review history of their article (what does this mean?). If published, this will include your full peer review and any attached files.

Reviewer #1: No

Reviewer #2: No

Reviewer #3: No

---

## [Author Response · Author response to Decision Letter 0]

3 Jun 2021

Reviewer: 1

Dear Reviewer,

Thank you for sending us your comments and suggestions. We are pleased to read, that you find our algorithm interesting. We have also prepared answers to your comments:

“1. The algorithm sections are quite difficult to read. The notations are very complex. It feels like each set of equations were using very non-overlapping sets of symbols that represent different things. It is true that different sets of equations described different steps/components of the algorithm. If the notations can be significantly simplified, it will be a lot easier for readers to follow this paper and appreciate the algorithmic designs.”

Answer: 

We carefully revised parts of our notation to avoid confusion due to the naming of indices and made the naming of variables which are optimized more consistent. The symbol t now always indicates the time. Previously t was used for source (t+) & sink (t⁻) nodes and for time points. This allowed us to replace the variable k used in the node indices (now always u*,t instead of u*,k) and to avoid confusions with node naming, now referred to as source (q+) and sink (q-). The notation for optimization variables is now more consistent. The flow variables are renamed to zf (f before) and coupled flow variables to zfjl (fjl.before) The optimization variables in the untangling problem are now: zepn (eij before), zrm (ml before), and zsn (si before). We also adapted the notation of the cost functions for the untangling problem: cepn (c(eij) before), cmr (c(ml) before), and csn (c(si) before). Moreover, we renamed some indices to avoid confusion between the indices of the two different optimization problems.

„2. The figures are quite repetitive in some sense, the same graph topology appearing in 4 different figures. Again, I understand that they convey different information on different steps of the algorithm, but I would be better to consolidate them into one figure with multiple sub-figures, so that it is easier for readers to understand the whole flow. “

Answer:

We agree that the overall number of figures can be reduced to improve the flow for the readers. We adapted the figures accordingly – We kept Figure 1 for conceptual overview. Figures 2&3 are combined so the graph example and the costs are shown in a single figure. We combined Figures 4&5 now showing the extracted features and how they can be used to match correctly and erroneously segmented objects. Figures 5&6 are combined as well, showing the untangling operations and the untangling problem.

Yours sincerely,

Katharina Löffler

Reviewer: 2

Dear Reviewer,

We are excited to read that you think our paper, which proposes a cell tracking algorithm, has merit to be published in PLOS ONE. We have also prepared answers to your comments:

“1. The algorithms seems to work on separated cell (cell lines only), and not on tissues (incompletely acquired objects). This should be discussed.”

Answer: 

Usually the problem on tissues is more a segmentation issue, as these cells are rather constrained in their position. Hence, most tissue analysis problems (especially in pathology) do not require a tracking analysis. In contrast, on cell data sets where cells can move freely the tracking is more challenging as now the tracking algorithm needs to handle large shifts in the cell position as well as segmentation errors, e.g. due to low resolution in the z-axis. However, our tracking algorithm can handle a variety of cell types (shapes, sizes and textures) and imaging acquisition techniques as shown in the added Table 3. The main prerequisite is a segmentation of the objects to track. 

“2. What kind of principle segmentation algorithm has been used (dynamic, static, Otsu, …) etc. or texture / transformation based methods (Fourier, Laplace, …)?”

Answer: 

In our initial manuscript we investigated how tracking algorithms handle erroneous segmentation data. Therefore, we simulated segmentation errors based on a provided, perfect ground truth. As the data sets Fluo-N2DH-SIM+ and Fluo-N3DH-SIM+ from the Cell Tracking Challenge are synthetic data sets, their ground truth segmentation masks are generated synthetically as well. Thus, no segmentation algorithm was needed. Now, we have added an evaluation of our tracking algorithm on a total of 15 data sets of the Cell Tracking Challenge, where we used a deep learning based segmentation approach. The new Table 3 shows that such a combination is very competitive. However, any segmentation approach could be combined with our proposed tracking step.

“3. Are different cell types included in this study?”

Answer:

We extended our analysis and applied the tracking algorithm on a total of 15 different cell data sets, showing different cell types imaged in 2D or 3D as well as different imaging conditions – Fluorescently counterstained, Bright Field, Phase Contrast, and Differential Interference Contrast. The results are shown in Table 3. The discussion has been adapted accordingly.

“4. The authors should clearly distinguish the different mandatory compartments of the algorithms, namely (image acquisition (including image quality), object detection (differentiation potential Object / background), and object identification (object features) and classification (object class) for additional analysis (for example texture).

For example see and cite) (KAYSER, Klaus; BORKENFELD, Stephan; KAYSER, Gian. Digital Image Content and Context Information in Tissue-based Diagnosis. Diagnostic Pathology, [S.l.], v. 4, n. 1, dec. 2018. ISSN 2364-4893; GÖRTLER, Jürgen et al. Cognitive Algorithms and digitized Tissue – based Diagnosis. Diagnostic Pathology, [S.l.], v. 3, n. 1, july 2017. ISSN 2364-4893.“

Answer: 

As we propose the tracking step for a tracking by detection pipeline, a segmentation method – for example deep learning based – needs to be selected by the user. We agree that our tracking algorithm has to be embedded into more complex analysis pipelines, but we think that a detailed discussion of complete pipelines is beyond the scope of the paper.

“5. A few words should be addressed to future development (understanding) how structures 8cells) and functions (movements) might depend on each other. For example see and cite (Kayser, Klaus, Borkenfeld, Stephan, Fang, Wei-Kleiner, Kayser, Gian: Digital Pathology Where did – do – will you go? Content and Context Analysis of Communication in Digital Pathology. https://doi.org/10.17629/www.medical-journal-of-virtual-science.de-2021-1:2.“

Answer: 

Thank you for suggesting this interesting paper. In our opinion, it is definitely a future task to correlate structure and behavior. We assume, however, that pathology is here a very ambitious field because most samples cannot be observed over time by an in vivo imaging approach. While we see a clear motivation for segmentation approaches on pathological images, we find it challenging to see the connection between the suggested paper and our proposed cell tracking method, as usually pathological images capture immobile cells. Thus, we have decided not to cite it.

Yours sincerely,

Katharina Löffler

 

Reviewer: 3

Dear Reviewer,

Thank you for your comments and suggestions. We are pleased to read that you appreciated the overall presentation of the content. We have also prepared answers to your suggestions:

“The drawback of the presentation is that the algorithm was only tested on two of 19 available datasets from the CTC. Another disadvantage is that only the tracking accuracies TRA are presented. With this minimal comparison, the evaluation of the proposed new algorithm cannot be performed satisfactorily.

Therefore, I would suggest adding further samples and adding the values for segmentation accuracies, detection accuracies, cell segmentation benchmarks and cell tracking benchmarks.“ 

Answer:

We extended our initial analysis on synthetically degraded segmentation data by computing the DET and SEG scores from the CTC as well. We have updated the corresponding figures. In addition, we added our results from the Cell Tracking Benchmark on this year’s CTC, where we participated on a diverse set of 2D and 3D data sets (15 in total). To evaluate the performance of our algorithm, we used the same manually tunable parameter setting on all submitted 2D and 3D data sets. We think the added results emphasize the capabilities of our tracking algorithm to perform competitively on a diverse set of 2D and 3D data sets without the need of fine tuning.

Yours sincerely,

Katharina Löffler

---

## [Decision Letter · Decision Letter 1]

14 Jun 2021

PONE-D-21-08192R1

A graph-based cell tracking algorithm with few manually tunable parameters and automated segmentation error correction

PLOS ONE

Dear Dr. Löffler,

Thank you for submitting your manuscript to PLOS ONE. After careful consideration, we feel that it has merit but does not fully meet PLOS ONE’s publication criteria as it currently stands. Therefore, we invite you to submit a revised version of the manuscript that addresses the points raised during the review process.

I fully agree with the comments of one reviewer. Please make corrections according to the suggestions.  

We look forward to receiving your revised manuscript.

Kind regards,

Konradin Metze

Academic Editor

PLOS ONE

Reviewers' comments:

Reviewer's Responses to Questions

**Comments to the Author**

1. If the authors have adequately addressed your comments raised in a previous round of review and you feel that this manuscript is now acceptable for publication, you may indicate that here to bypass the “Comments to the Author” section, enter your conflict of interest statement in the “Confidential to Editor” section, and submit your "Accept" recommendation.

Reviewer #1: All comments have been addressed

Reviewer #2: All comments have been addressed

Reviewer #3: All comments have been addressed

2. Is the manuscript technically sound, and do the data support the conclusions?

Reviewer #1: Yes

Reviewer #2: Partly

Reviewer #3: Yes

3. Has the statistical analysis been performed appropriately and rigorously? 

Reviewer #1: N/A

Reviewer #2: Yes

Reviewer #3: Yes

4. Have the authors made all data underlying the findings in their manuscript fully available?

Reviewer #1: Yes

Reviewer #2: Yes

Reviewer #3: Yes

5. Is the manuscript presented in an intelligible fashion and written in standard English?

Reviewer #1: Yes

Reviewer #2: Yes

Reviewer #3: Yes

6. Review Comments to the Author

Reviewer #1: The authors have addressed all my comments. With simpler notations and consolidated figures, the manuscript now is easier to follow compared to the previous submission.

Reviewer #2: The authors have responded to all comments of the reviewer; however, they are not willing to accept the required modifications / recommendations. There are some major inconsistencies, for example 'ground truth does not need segmentation' which need essential explanation.

Therefore, I cannot recomment the article for publication.

Reviewer #3: (No Response)

7. PLOS authors have the option to publish the peer review history of their article (what does this mean?). If published, this will include your full peer review and any attached files.

Reviewer #1: No

Reviewer #2: No

Reviewer #3: **Yes: **Helmut Ahammer

---

## [Author Response · Author response to Decision Letter 1]

17 Jun 2021

Dear Konradin Metze,

thank you very much for editing our revised manuscript entitled “A graph-based cell tracking algorithm with few manually tunable parameters and automated segmentation error correction”. We assume that our background and the one of reviewer 2 are quite different, resulting in an unfortunate misunderstanding on our side on how to interpret the requested adaptions. We hope to clarify all misunderstanding by adding further explanations and references to the paper. The changes to the compared to the last revision are highlighted in blue (added) and red (removed) in the tracked changes manuscript version.

Yours sincerely,

Katharina Löffler

Reviewer: 1

Dear Reviewer,

Thank you for your helpful suggestions to improve the notation and graphics.

Yours sincerely,

Katharina Löffler

Reviewer: 2

Dear Reviewer,

We are sorry to read that you feel we are unwilling to accept your suggested modifications. We might have misunderstood parts of your suggestions and thought that you just wanted some further explanations, which we gave to you. However, we see that the paper also needed some further editing. We hope that the added comments and modifications on the paper can help to resolve this misunderstanding. Please find below answers to your comments.

-----

We think the first point 1 from your first revision was due to a misunderstanding of the scope and the content of the paper, which we decided to clarify in the rebuttal letter only:

“1. The algorithms seems to work on separated cell (cell lines only), and not on tissues (incompletely acquired objects). This should be discussed.”

Answer:

As we proposed a tracking algorithm and not a segmentation algorithm our focus lies on data sets showing moving cells. Therefore, we used data sets from the in the cell tracking community well-established Cell Tracking Challenge to evaluate and compare our cell tracking algorithm. 

Unfortunately, we have trouble understanding on what is meant by incompletely acquired objects in the scope of a tracking problem.

-----

All other suggestions from the first revision have now been added to the paper (or were already resolved but not clearly indicated in our rebuttal letter). We would like to summarize how we addressed your required modifications / recommendation by providing quotes from the paper:

“2. What kind of principle segmentation algorithm has been used (dynamic, static, Otsu, …) etc. or texture / transformation based methods (Fourier, Laplace, …)?”

Answer:

We now added the text passage (page 3, lines 81-84 in the tracked changes manuscript version)

“In this section, we describe our tracking algorithm, which is able to process 2D and 3D image sequences, in more detail. To create a tracking by detection algorithm, our proposed tracking algorithm can be combined with an arbitrary segmentation algorithm, which predicts instance segmentation masks.”

to clarify that our tracking algorithm can be combined with any segmentation approach.

Moreover, for our participation on the 6th edition of the CTC (results in Table 3) we selected a deep learning based segmentation (page 16, lines 541-543 in the tracked changes manuscript version) 

“For segmentation, we chose a deep learning based segmentation approach which utilizes cell and neighbor distances [9, 57]. The derived segmentation masks and the raw image sequence were fed into our tracking algorithm. ”

Since we added this text passage already in the last revision and only changes to that revision are highlighted in blue, that change is not highlighted.

For our analysis on how different tracking algorithms behave, when provided with erroneous segmentation data, we used the perfect, segmentation masks from the data sets Fluo-N2DH-SIM+ and Fluo-N3DH-SIM+ which are provided with the data. These two data sets are synthetically generated; thus no additional segmentation is needed to generate the used ground truth segmentation masks. We emphasize that these two data sets are synthetic by adding (page 13, lines 429-430 in the tracked changes manuscript version)

“Both cell data sets show synthetically generated human leukemia cells, where Fluo-N2DH-SIM+ is a 2D data set and Fluo-N3DH-SIM+ is a 3D data set.”

-----

“3. Are different cell types included in this study?”

Answer:

We evaluated on 15 different cell data sets, please see Table 3 (page 17) for the different examined cell types. We added the table already in the last revision to answer your question.

-----

“4. The authors should clearly distinguish the different mandatory compartments of the algorithms, namely (image acquisition (including image quality), object detection (differentiation potential Object / background), and object identification (object features) and classification (object class) for additional analysis (for example texture).

For example see and cite) (KAYSER, Klaus; BORKENFELD, Stephan; KAYSER, Gian. Digital Image Content and Context Information in Tissue-based Diagnosis. Diagnostic Pathology, [S.l.], v. 4, n. 1, dec. 2018. ISSN 2364-4893; GÖRTLER, Jürgen et al. Cognitive Algorithms and digitized Tissue – based Diagnosis. Diagnostic Pathology, [S.l.], v. 3, n. 1, july 2017. ISSN 2364-4893.“

Answer:

We added a paragraph to clarify the additional components needed to create a tracking by detection approach. Furthermore, we emphasized that the tracking actually is a step in an image analysis pipeline (page 3, lines 81-86 in the tracked changes manuscript version):

“In this section, we describe our tracking algorithm, which is able to process 2D and 3D image sequences, in more detail. To create a tracking by detection algorithm, our proposed tracking algorithm can be combined with an arbitrary segmentation algorithm, which predicts instance segmentation masks. Moreover, the tracking can be included in a full image analysis pipeline which typically consists of sample preparation and imaging, cell segmentation, cell tracking, and subsequent analysis [51, 52].”

Therefore, we added also the new references:

[51] Megason SG, Fraser SE. Imaging in systems biology. Cell. 2007;130(5):784–795.doi:10.1016/j.cell.2007.08.031.52.

[52] Eliceiri KW et al. Biological imaging software tools. Nature Methods. 2012;9(7):697–710.doi:10.1038/nmeth.2084

-----

“5. A few words should be addressed to future development (understanding) how structures 8cells) and functions (movements) might depend on each other. For example see and cite (Kayser, Klaus, Borkenfeld, Stephan, Fang, Wei-Kleiner, Kayser, Gian: Digital Pathology Where did – do – will you go? Content and Context Analysis of Communication in Digital Pathology. https://doi.org/10.17629/www.medical-journal-of-virtual-science.de-2021-1:2.“

Answer:

To put our work into the greater picture, we added a paragraph to the conclusion (page 19, lines 618-622 in the tracked changes manuscript version):

“We envision that the steady improvement of automated cell tracking approaches concerning the accuracy, run time, and ease of applicability will lead to powerful tools to analyze cell behavior quantitatively. The derived insights on the cell behavior can then help to deepen our understanding of the mechanisms influencing cell migration or, for instance, how cell migration and the formation of structures depend on each other.”

-----

Concerning the last point of your second review:

“There are some major inconsistencies, for example 'ground truth does not need segmentation' which need essential explanation.

Therefore, I cannot recomment the article for publication”

Answer:

We are highly motivated to further improve the quality of our paper and therefore are looking forward to your feedback on major inconsistencies. Unfortunately, as only one of them is provided here we are only able to resolve this one.

We would like to refer to (page13, lines 426-428 of the tracked changes manuscript version):

“We select the cell data sets Fluo-N2DH-SIM+ and Fluo-N3DH-SIM+ from the CTC [5, 50] for evaluation, as they are publicly available and provide a fully annotated ground truth, i.e. segmentation masks are given for all cells as well as the cell lineage.” 

In which we explain what kind of ground truth data is available; all segmentation masks as well as the lineage are provided by the two selected CTC data set. As the data sets are simulated, no additional segmentation algorithm is needed. We tried to emphasize the synthetic nature of the data set by making the following adaption (page 13, lines 429-430 in the tracked changes manuscript version):

“Both cell data sets show synthetically generated human leukemia cells, where Fluo-N2DH-SIM+ is a 2D data set and Fluo-N3DH-SIM+ is a 3D data set.”

To evaluate how different tracking algorithms handle different types of segmentation errors, we then modify these provided segmentation masks to model segmentation errors (page 14, lines 433-456 of the tracked changes manuscript version). 

We hope, that in the new manuscript there is no more room for confusion. Thank you for your helpful suggestions to improve this manuscript.

Yours sincerely,

Katharina Löffler

------------------

Reviewer 3:

Dear Reviewer,

Thank you for your suggestions to include additional metrics and strengthening the results section by evaluating on more data sets.

Yours sincerely,

Katharina Löffler

---

## [Decision Letter · Decision Letter 2]

16 Jul 2021

PONE-D-21-08192R2

A graph-based cell tracking algorithm with few manually tunable parameters and automated segmentation error correction

PLOS ONE

Dear Dr. Löffler,

Thank you for submitting your re-revised manuscript to PLOS ONE. After careful consideration, we feel that it has merit but does not fully meet PLOS ONE’s publication criteria as it currently stands. Therefore, we invite you to submit a revised version of the manuscript that addresses the points raised during the review process.

The manuscript has been improved without any doubt. But there are still some questions open, as reviewer 2 is pointing out. I agree with these comments and think that all these topics  must be discussed in detail and included into the paper, in order to mainatain a high scientific standard.

We look forward to receiving your revised manuscript.

Kind regards,

Konradin Metze

Academic Editor

PLOS ONE

Journal Requirements:

Additional Editor Comments (if provided):

Reviewers' comments:

Reviewer's Responses to Questions

**Comments to the Author**

1. If the authors have adequately addressed your comments raised in a previous round of review and you feel that this manuscript is now acceptable for publication, you may indicate that here to bypass the “Comments to the Author” section, enter your conflict of interest statement in the “Confidential to Editor” section, and submit your "Accept" recommendation.

Reviewer #1: All comments have been addressed

Reviewer #2: (No Response)

Reviewer #3: All comments have been addressed

2. Is the manuscript technically sound, and do the data support the conclusions?

Reviewer #1: Yes

Reviewer #2: Partly

Reviewer #3: Yes

3. Has the statistical analysis been performed appropriately and rigorously? 

Reviewer #1: N/A

Reviewer #2: Yes

Reviewer #3: Yes

4. Have the authors made all data underlying the findings in their manuscript fully available?

Reviewer #1: Yes

Reviewer #2: Yes

Reviewer #3: Yes

5. Is the manuscript presented in an intelligible fashion and written in standard English?

Reviewer #1: Yes

Reviewer #2: Yes

Reviewer #3: Yes

6. Review Comments to the Author

Reviewer #1: The authors have sufficiently addressed my comments in the previous revision. This new round of revision also looks good to me.

Reviewer #2: There remains still a principle problem, which the authors should clarify: They state, that their tracking algorithm is not dependent upon the segmentation algorithm. That is only true, if the background remains the same during the tracking process, an assumption that is quite unlikely in reality. Examples are, if additional artifacts appear or if the tracked individuals confuse, or if the distance between different objects shrinks to zero? This is a major problem in graph theory too; its application requires a 'none-zero distance' between different nodes (or non-zero edges). Therefore, I propose again to discuss briefly the aspect of image quality and its impact on any derived action and to cite the given references. Herein I repeat my opinion. I cannot propose an article for publication which disregards important practical and theoretical aspects.

Reviewer #3: (No Response)

7. PLOS authors have the option to publish the peer review history of their article (what does this mean?). If published, this will include your full peer review and any attached files.

Reviewer #1: No

Reviewer #2: No

Reviewer #3: **Yes: **Helmut Ahammer

---

## [Author Response · Author response to Decision Letter 2]

22 Jul 2021

Dear Konradin Metze,

thank you very much for your patience and time in editing our revised manuscript entitled “A graph-based cell tracking algorithm with few manually tunable parameters and automated segmentation error correction”. We hope to clarify the remaining concerns by discussing the influence of image quality and temporal resolution on the segmentation and tracking performance in the discussion section and added further references. The changes compared to the last revision are highlighted in blue (added) in the tracked changes manuscript version.

Yours sincerely,

Katharina Löffler

Reviewer: 2

Dear Reviewer,

Thank you very much for your patience and time in reviewing our manuscript. Please find below answers to your comments.

“There remains still a principle problem, which the authors should clarify: They state, that their tracking algorithm is not dependent upon the segmentation algorithm. That is only true, if the background remains the same during the tracking process, an assumption that is quite unlikely in reality. Examples are, if additional artifacts appear or if the tracked individuals confuse, or if the distance between different objects shrinks to zero? This is a major problem in graph theory too; its application requires a 'none-zero distance' between different nodes (or non-zero edges). Therefore, I propose again to discuss briefly the aspect of image quality and its impact on any derived action and to cite the given references. Herein I repeat my opinion. I cannot propose an article for publication which disregards important practical and theoretical aspects.”

Answer:

To discuss the dependence of the segmentation and tracking performance on the image quality, we added the following paragraph [pages 18-19 lines 596-606]:

“We would like to emphasize that the tracking performance depends on the instance segmentation which in turn depends on the image quality. To reduce the dependence on the image quality, the image quality can be improved substantially by applying image restoration methods before segmentation [58,59]. In addition, instance segmentation approaches applicable to a broad range of imaging conditions exist [60]. While our tracking approach can be combined with an arbitrary instance segmentation approach, different instance segmentation approaches can be prone to different types and quantities of segmentation errors. Our results on simulated, erroneous segmentation data show, that our tracking algorithm can correct certain types of randomly occurring segmentation errors, however, with decreasing segmentation quality the tracking quality decreases as well. “

Moreover, we now discussed the influence of the temporal resolution on the tracking performance as well as sketched a limitation of our tracking algorithm [page 19 lines 607-618]:

“In general, the tracking performance also depends on the temporal resolution of the image sequence. If the temporal resolution is high with respect to the cell movements -- cell movements are small between frames with respect to the cell size -- simple, nearest neighbor assignment is sufficient [61]. However, when the temporal resolution is restricted, e.g. to avoid photodamage, large cell movements between successive frames are possible. To assign the segmented cells correctly, more advanced approaches, such as graph-based approaches, are needed. As the results of the CTC show, our position-based costs perform well on a broad set of real world cell data sets, however, there are scenarios which will result in wrong assignments. For instance, consider two cells at time point t which have swapped their positions at time t+1, which is impossible to detect using position-based costs. To resolve such cases, the tracking costs can be adapted using more complex features based on texture or morphology of single cells [25,30,37].”

To support the dependence of segmentation and tracking performance on the image quality and the dependence of the tracking on the temporal resolution of the image sequence we added the following references:

[58] Weigert M, Schmidt U, Boothe T, Müller A, Dibrov A, Jain A, et al. Content-aware image restoration: Pushing the limits of fluorescence microscopy. Nature Methods. 2018;15(12):1090–1097. doi:10.1038/s41592-018-0216-7.

[59] Belthangady C, Royer LA. Applications, promises, and pitfalls of deep learning for fluorescence image reconstruction. Nature Methods. 2019;16(12):1215–1225.doi:10.1038/s41592-019-0458-z.

[60] Caicedo JC, Goodman A, Karhohs KW, Cimini BA, Ackerman J, Haghighi M, et al. Nucleus segmentation across imaging experiments: the 2018 Data Science Bowl. Nature Methods. 2019;16(12):1247–1253. doi:10.1038/s41592-019-0612-7.

[61] Yang FW, Tomášová L, Guttenberg Zv, Chen K, Madzvamuse A. Investigating optimal time step intervals of imaging for data quality through a novel fully-automated cell tracking approach. Journal of Imaging. 2020;6(7). doi:10.3390/jimaging6070066.

We hope the added paragraphs and references resolve the remaining remarks.

Yours sincerely,

Katharina Löffler

------------------

---

## [Decision Letter · Decision Letter 3]

17 Aug 2021

A graph-based cell tracking algorithm with few manually tunable parameters and automated segmentation error correction

PONE-D-21-08192R3

Dear Dr. Löffler,

We’re pleased to inform you that your manuscript has been judged scientifically suitable for publication and will be formally accepted for publication once it meets all outstanding technical requirements.

Kind regards,

Konradin Metze

Academic Editor

PLOS ONE

Additional Editor Comments (optional):

Reviewers' comments:

Reviewer's Responses to Questions

**Comments to the Author**

1. If the authors have adequately addressed your comments raised in a previous round of review and you feel that this manuscript is now acceptable for publication, you may indicate that here to bypass the “Comments to the Author” section, enter your conflict of interest statement in the “Confidential to Editor” section, and submit your "Accept" recommendation.

Reviewer #2: (No Response)

Reviewer #4: All comments have been addressed

2. Is the manuscript technically sound, and do the data support the conclusions?

Reviewer #2: Partly

Reviewer #4: Yes

3. Has the statistical analysis been performed appropriately and rigorously? 

Reviewer #2: Yes

Reviewer #4: Yes

4. Have the authors made all data underlying the findings in their manuscript fully available?

Reviewer #2: Yes

Reviewer #4: Yes

5. Is the manuscript presented in an intelligible fashion and written in standard English?

Reviewer #2: Yes

Reviewer #4: Yes

6. Review Comments to the Author

Reviewer #2: Unfortunately, the authors have not fulfilled the recommendations of the reviewer. They just added some fluorescence specificities and limited the basic problems of their algorithms to spatial resolution. There is no indication that the proposed algorithm will work with usual cytological and histologic color images. I am afraid that they are not aware of the real problem of object / background distinction in combination with the observation period. Therefore, the quality of this article does not reach the required level of the journal in this third revision.

Reviewer #4: The authors propose a method for cell tracking with a reduced number of tunable parameters and automatic segmentation error correction. Their proposal is validated on the well established CTC benchmark database. The manuscript is well written and the methodology is innovative in this type of task and the achieved results are competitive with the state-of-the-art. In this scenario, I recommend acceptance.

7. PLOS authors have the option to publish the peer review history of their article (what does this mean?). If published, this will include your full peer review and any attached files.

Reviewer #2: No

Reviewer #4: No

---

## [Editor Report · Acceptance letter]

27 Aug 2021

PONE-D-21-08192R3 

A graph-based cell tracking algorithm with few manually tunable parameters and automated segmentation error correction 

Dear Dr. Löffler:

I'm pleased to inform you that your manuscript has been deemed suitable for publication in PLOS ONE. Congratulations! Your manuscript is now with our production department. 

Kind regards, 

on behalf of

Prof. Konradin Metze 

Academic Editor

PLOS ONE